# Developmental deprivation-induced perceptual and cortical processing deficits in awake-behaving animals

**Justin D Yao[1]\*, Dan H Sanes[1,2,3,4]\***

[1]Center for Neural Science, New York University, New York, United States;
[2]Department of Psychology, New York University, New York, United States;
[3]Department of Biology, New York University, New York, United States;
[4]Neuroscience Institute, NYU Langone Medical Center, New York, United States

**Abstract** Sensory deprivation during development induces lifelong changes to central nervous system function that are associated with perceptual impairments. However, the relationship between neural and behavioral deficits is uncertain due to a lack of simultaneous measurements during task performance. Therefore, we telemetrically recorded from auditory cortex neurons in gerbils reared with developmental conductive hearing loss as they performed an auditory task in which rapid fluctuations in amplitude are detected. These data were compared to a measure of auditory brainstem temporal processing from each animal. We found that developmental HL diminished behavioral performance, but did not alter brainstem temporal processing. However, the simultaneous assessment of neural and behavioral processing revealed that perceptual deficits were associated with a degraded cortical population code that could be explained by greater trial-to-trial response variability. Our findings suggest that the perceptual limitations that attend early hearing loss are best explained by an encoding deficit in auditory cortex.
DOI: https://doi.org/10.7554/eLife.33891.001

\*For correspondence:
jdyao@nyu.edu (JDY);
dhs1@nyu.edu (DHS)

**Competing interests:** The authors declare that no competing interests exist.

## Introduction

Many forms of developmental deprivation, including the loss of hearing or vision, induce lifelong behavioral deficits that are associated with degraded central nervous system (CNS) function (*Kiorpes, 2006*; *Petersen, 2007*; *Kalinichev and Francis, 2010*; *Sanes and Woolley, 2011*; *Espinosa and Stryker, 2012*). However, the precise relationship between perception and sensory coding following sensory deprivation is uncertain because measurements are generally obtained from separate groups of animals. This uncertainty can be attributed, in part, to the dramatic differences of sensory-evoked responses elicited during task engagement versus disengagement (*Hubel et al., 1959*; *Burton et al., 1997*; *Spitzer et al., 1988*; *McAdams and Maunsell, 1999a*; *1999b*; *Treue and Maunsell, 1999*; *Reynolds et al., 2000*; *Steinmetz et al., 2000*; *Fritz et al., 2003*; *2005*; *Chapman and Meftah, 2005*; *Elhilali et al., 2007*; *Cohen and Newsome, 2008*; *Cohen and Maunsell, 2009*; *Otazu et al., 2009*; *Lee and Middlebrooks, 2011*; *David et al., 2012*; *Niwa et al., 2015*; *Buran et al., 2014b*; *Schneider et al., 2014*; *Zhou et al., 2014*; *McGinley et al., 2015*; *Slee and David, 2015*; *von Trapp et al., 2016*; *Carcea et al., 2017*; *Caras and Sanes, 2017*). One recent advancement has been to examine perceptual and sensory encoding in the same animals following developmental conductive hearing loss (HL) (*Keating et al., 2013*; *2015*), but the direct relationship between neural dysfunction and perceptual skills can only be inferred without obtaining simultaneous neural and psychophysical measurements from the same animals. A related question concerns the central location(s) of HL-induced degraded neural processing that might explain perceptual deficits. Since permanent HL induced during early development can impair neural encoding

mechanisms throughout the auditory pathway, many candidate mechanisms have been suggested (*Knudsen et al., 1982*; *1984a*; *1984b*; *Mogdans and Knudsen, 1993*; *Mogdans and Knudsen, 1994*; *Raggio and Schreiner, 1999*; *Snyder et al., 2000*; *DeBello et al., 2001*; *Moore et al., 1999*; *Yu et al., 2005*; *Takahashi et al., 2006*; *Fallon et al., 2008*; *Razak et al., 2008*; *Popescu and Polley, 2010*; *Rosen et al., 2012*; *Polley et al., 2013*). To address these issues, we recorded from ACx in HL-reared and control (Ctl) animals as they performed an auditory detection task, and compared behavioral sensitivity to ACx encoding and auditory brainstem processing.

When children are deprived of normal auditory input early in life, they display permanent CNS changes and diminished perceptual sensitivity to the spectral and temporal features that comprise natural sounds (*Ponton and Eggermont, 2001*; *Sharma et al., 2007*; *Gilley et al., 2008*; *Park et al., 2015*; *Landsberger et al., 2018*). One set of features, temporal fluctuations in sound level, support speech comprehension, and perceptions of rhythm, prosody, musical attack, and pitch (*Burns and Viemeister, 1976*; *Singh and Theunissen, 2003*). Despite the importance of these amplitude modulation (AM) cues for aural communication, perceptual sensitivity to these signals displays a relatively prolonged period of developmental maturation, with detection improving into adolescence (*Hall and Grose, 1994*; *Sarro and Sanes, 2010*; *Banai et al., 2011*). This prolonged maturation suggests that temporal processing could be vulnerable to environmental experience, including HL. In fact, there is clinical and experimental evidence demonstrating that developmental HL impairs AM sensitivity and discriminability, particularly for fast (>100 Hz) modulation rates (*Bacon and Viemeister, 1985*; *Grant et al., 1998*; *Park et al., 2015*). Recent studies have shown that developmental HL, which can occur in children with otitis media (*Gravel and Wallace, 2000*; *Whitton and Polley, 2011*), leads to long-lasting cellular deficits in auditory cortex (ACx) (*Xu et al., 2007*; *Takesian et al., 2012*; *Mowery et al., 2015*; *2016*; *2017*). Therefore, it is plausible that a HL-induced ACx encoding deficit could explain, in part, degraded behavioral performance (*Rosen et al., 2012*; *Buran et al., 2014a*; *Caras and Sanes, 2015*; *von Trapp et al., 2017*; *Ihlefeld et al., 2016*; *Green et al., 2017*).

We tested the hypothesis that developmental conductive HL disrupts ACx encoding of fast AM rates, thereby degrading perceptual sensitivity. Since functional impairments emerge in both subcortical and cortical structures following early deprivation (*Sanes, 2013*; *Butler and Lomber, 2013*), our primary aim was to determine whether neural processing deficits in primary sensory cortex are associated with diminished perceptual performance. Our findings suggest that the neural correlates of HL-induced perceptual deficits do not emerge at the auditory brainstem, but are associated with cortical population-level activity that could be read out by decoding regions downstream of ACx.

## Results

### Developmental conductive hearing loss impairs behavioral sensitivity to AM rate

To determine whether animals reared with permanent bilateral conductive HL display a reduction in perceptual sensitivity to fast AM rates, all animals were trained to perform a Go-Nogo AM noise detection task at rates of 64, 128, 256, and 512 Hz ('Go' depths: −15 to 0 dB rel. 100%; 'Nogo' depth: unmodulated noise). Animals were subsequently tested at each AM rate until they reached asymptotic performance where threshold values remained within ±3 dB across a minimum of 3 sessions. *Figure 1A* displays example psychometric functions at the four tested AM rates from single testing sessions for representative Ctl (black) and HL (orange) animals. AM detection thresholds ($d'$=1) were calculated from individual psychometric sessions and averaged at each AM rate to construct temporal modulation transfer functions (TMTFs; *Viemeister, 1979*) for Ctl and HL groups (*Figure 1B*). AM thresholds displayed a significant interaction between hearing status and AM rate (two-way mixed model ANOVA; $F_{(3,48)}$ = 30.3, p<0.0001). To determine whether detection thresholds differed between groups at specific AM rates, we used post-hoc two-tailed t-tests (Holm-Bonferroni-corrected). Thresholds from HL animals were similar to Ctl animals at the slowest AM rate tested, 64 Hz (p>0.05, t = 2.07), but were markedly worse at 128 Hz (p<0.05, t = 2.75) and 256 Hz (p<0.0001, t = 9.41), and displayed the greatest deficit at 512 Hz (p<0.0001, t = 12.6).

To further characterize the differences in psychometric performance, each animal's TMTF was fitted with an exponential function, using two fitting parameters to quantify separate aspects of the

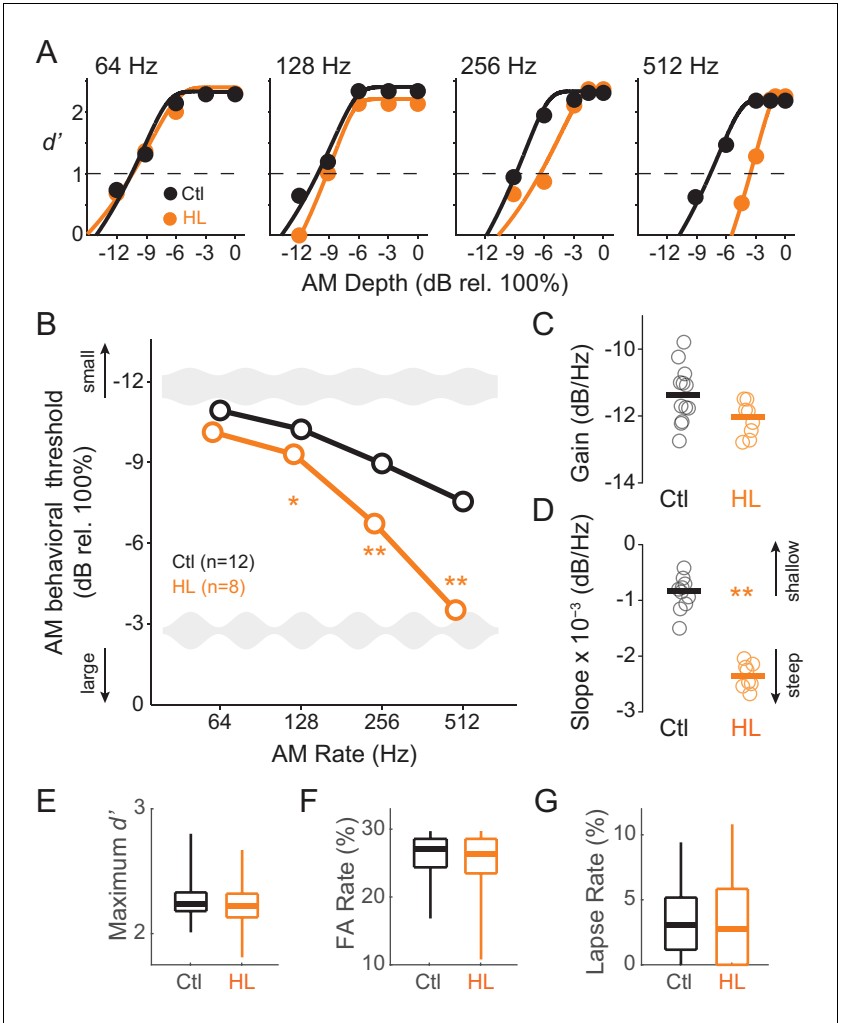

**Figure 1.** Developmental conductive hearing loss impairs AM rate detection. (**A**) Psychometric functions across four tested AM rates (64, 128, 256, and 512 Hz) from a single testing session for a representative Ctl (black) and HL (orange) animal. (**B**) Average temporal modulation transfer functions for Ctl and HL animals. Mean ±1 SEM. Error bars are smaller than symbol dimensions. (**C**) Distribution of gain values extracted from each animal's fitted exponential function. (**D**) Distribution of slope values extracted from each animal's fitted exponential function. (**E**) Boxplots of maximum $d'$ from all test sessions of Ctl and HL animals. (**F**) Boxplots of false alarm (FA) rate from all test sessions of Ctl and HL animals. (**G**) Boxplots of lapse rate from all test sessions of Ctl and HL animals. Thick horizontal bars represent median values. The boxplots span the 25th to 75th distribution percentiles, and the whiskers span the 5th and 95th distribution percentiles. *p<0.05; **p<0.0001.
DOI: https://doi.org/10.7554/eLife.33891.002

TMTF (see Materials and Methods). The 'gain' parameter, represented by the y-intercept of the fitted exponential function, describes overall 'efficiency' to detect AM, whereas the 'slope' parameter describes the degree to which AM detection is influenced by modulation rate. *Figure 1C and D* plot distributions of gain and slope coefficients, respectively. Although gain values were similar between Ctl and HL groups (two-sample Wilcoxon Rank Sum test; p>0.05, Z = 1.73), HL animals displayed steeper slopes (two-sample Wilcoxon Rank Sum test; p<0.0005, Z = 3.51). This suggests that while both Ctl and HL animals are equally sensitive to slow AM (64 Hz), HL animals possessed poorer temporal resolution across very fast rates.

Since HL animals performed poorly on fast AM rate detection, we examined other task performance variable that serve as proxies for nonsensory factors, such as attention or motivation. Specifically, no significant differences were observed for maximum $d'$ (two-sample Wilcoxon Rank Sum test;

Ctl median: 2.24 ± 0.23; HL median: 2.22 ± 0.25; p>0.05, Z = 1.91), false alarm rate (two-sample Wilcoxon Rank Sum test; Ctl median: 0.27 ± 0.04; HL median: 0.27 ± 0.06; p>0.05, Z = 1.31), or lapse rate (Ctl median: 3.08 ± 2.99%; HL median: 2.78 ± 3.48%; two-sample Wilcoxon Rank Sum; p>0.05, Z = 0.38). We also compared response latencies between Ctl versus HL animals for Go (approached water spout) and Nogo (repoked) conditions across each tested AM rate. No main effect of group (Ctl versus HL) was observed for trials when animals approached the water spout (two-way mixed model ANOVA; $F_{(1,5266)}$ = 3.97, p>0.05) or for trials when animals repoked (two-way mixed model ANOVA; $F_{(1,2514)}$ = 0.18, p>0.05). This suggests that there were no systematic behavioral differences in motor behavior between Ctl versus HL animals during psychometric testing. Together, these analyses suggest that poorer AM detection thresholds observed in the HL group (*Figure 1B*) was not due to a diminished ability to perform the task.

Although stimuli were designed to control for average power (*Viemeister, 1979*; *Wakefield and Viemeister, 1990*), we nonetheless used a subset of Ctl animals (n = 4) to test whether animals were responding to an average level cue. Specifically, we randomly varied all stimuli across a 12 dB range (45–57 dB SPL) within each session. If animals were monitoring average level, we would expect psychometric functions to vary across this stimulus dimension. In contrast, each individual animal displayed robust AM detection with *d'* values > 2, and performance did not vary with average sound level (*Figure 2*). Across the four AM rates tested, linear fit slopes did not significantly vary from 0 (range: 0.00–0.006; one-sample t-test; p>0.05, t = 0.17–1.78), indicating that the sensory decision was not based on stimulus level.

## Perceptual deficits were not correlated with brainstem processing

One possible basis for HL-induced perceptual deficits is degraded processing at or below the level of inferior colliculus. To examine this, we recorded auditory brainstem responses (ABR) from Ctl (n = 9) and HL (n = 8) animals that completed psychometric testing on the AM detection task (*Figure 1*). This permitted a direct comparison between perceptual and ABR measures. ABR thresholds to clicks, a general measure of brainstem function, were ~39 dB higher in HL versus Ctl animals (*Figure 3A*; two-sample t-test; $p<10^{-4}$, t = −15.48). This validates the different sound levels used

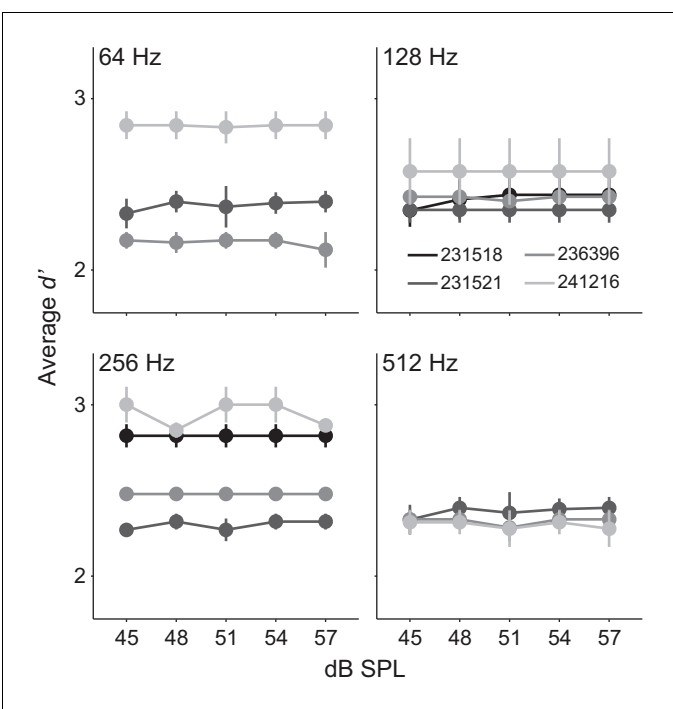

**Figure 2.** Gerbils did not use an average level cue. Average *d'* across 12 dB SPL sound levels from 4 Ctl animals tested at 64, 128, 256, and 512 Hz AM rates. Mean ± SEM.
DOI: https://doi.org/10.7554/eLife.33891.003

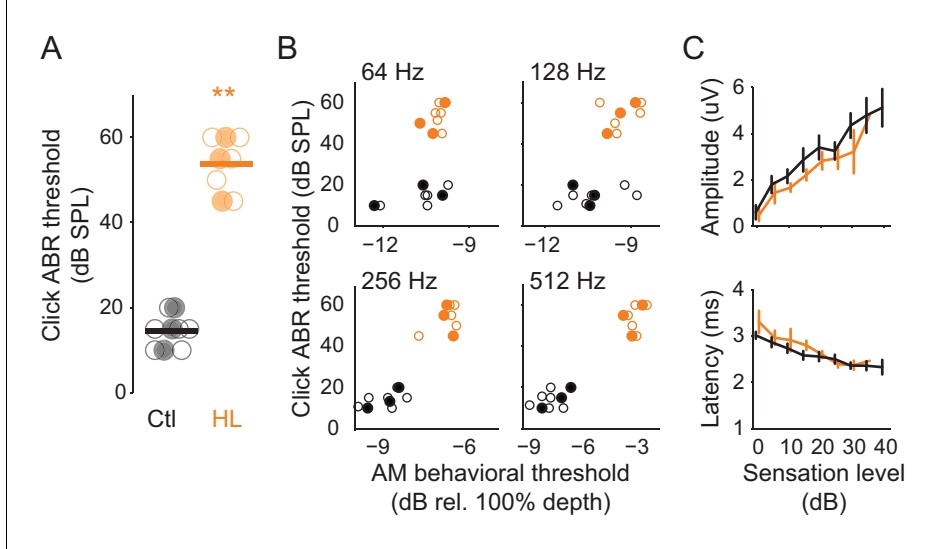

**Figure 3.** HL-related AM detection deficits cannot be explained by elevated ABR thresholds. (**A**) Click ABR thresholds from Ctl (black) and HL (orange) animals. ABR thresholds from HL animals were significantly greater than those from Ctl animals (**p<0.0001). Horizontal bars represent the group means. (**B**) Individual behavioral AM detection thresholds (x-axis) do not correlate with ABR click thresholds (y-axis). For HL animals, pearson r values ranged between 0.27–0.52, p>0.05. For Ctl animals, pearson r values ranged between 0.35–0.68, p>0.05. Filled symbols correspond to gerbils used for extracellular ACx recordings. (**C**) Mean ± SEM of maximum ABR amplitude (top) and latency (bottom) plotted as a function of sensation level (dB).
DOI: https://doi.org/10.7554/eLife.33891.004

for each hearing status group (Ctl = 45 or 50 dB SPL; HL = 90 dB SPL) during psychometric testing. To examine whether ABR thresholds were correlated with AM detection performance, we compared each animal's ABR click threshold to their average psychometric thresholds at each AM rate (*Figure 3B*). No significant correlations were observed for either the HL (Pearson's r = 0.27–0.52, p>0.05) or Ctl group (Pearson's r = 0.35–68, p>0.05). A comparison between ABR click and each animal's best AM detection threshold resulted in a similar outcome (HL, Pearson's r = 0.10–0.33, p>0.05; Ctl, Pearson's r = 0.48–0.53, p>0.05; data not shown). In addition, as illustrated in *Figure 3C* (top), ABR amplitudes grew significantly larger as sensation level increased (two-way mixed model ANOVA; $F_{(1,15)}$ = 127.6, p<0.0001), but there was no main effect of hearing status (two-way mixed model ANOVA; $F_{(1,15)}$ = 0.18, p>0.05), nor was there an interaction between sensation level and hearing status (two-way mixed model ANOVA; $F_{(6,90)}$ = 1.12, p>0.05). Furthermore, ABR peak latencies significantly decreased with increasing sensation level (two-way mixed model ANOVA; $F_{(1,15)}$ = 537.4, p<0.0001), but there was no main effect of hearing status (two-way mixed model ANOVA; $F_{(1,15)}$ = 0.74, p>0.05), nor was there an interaction between sensation level and hearing status (two-way mixed model ANOVA; $F_{(6,90)}$ = 1, p>0.05) (*Figure 3C*, bottom). Thus, elevated ABR click thresholds did not account for poorer AM sensitivity in the HL group, presumably because behavioral testing was performed at the same sensation level.

To assess brainstem processing of AM stimuli, we measured envelope following responses (EFRs) across the same AM rates and depths used for behavioral testing. As described in Materials and Methods, we used two recording configurations to obtain the best responses at 64–512 Hz (*Parthasarathy and Bartlett, 2012*; *Parthasarathy et al., 2014*). ABR traces are shown for electrode recording 'Configuration 1' (*Figure 4A*) and 'Configuration 2' (*Figure 4B*). Representative EFRs for 64 Hz (Configuration 1) and 512 Hz (Configuration 2) are shown across AM depth in *Figure 4C and D*, respectively. An FFT was performed on each trace and the peak amplitude within a ± 3 Hz window at the presented modulation rate was extracted to quantify the magnitude of phase-locking. We normalized these values across AM depth and fitted an exponential function to the corresponding data points (*Figure 4E,F*). EFR threshold was calculated as the lowest AM depth corresponding to an FFT amplitude that was twice as high as the driven amplitude during the unmodulated noise

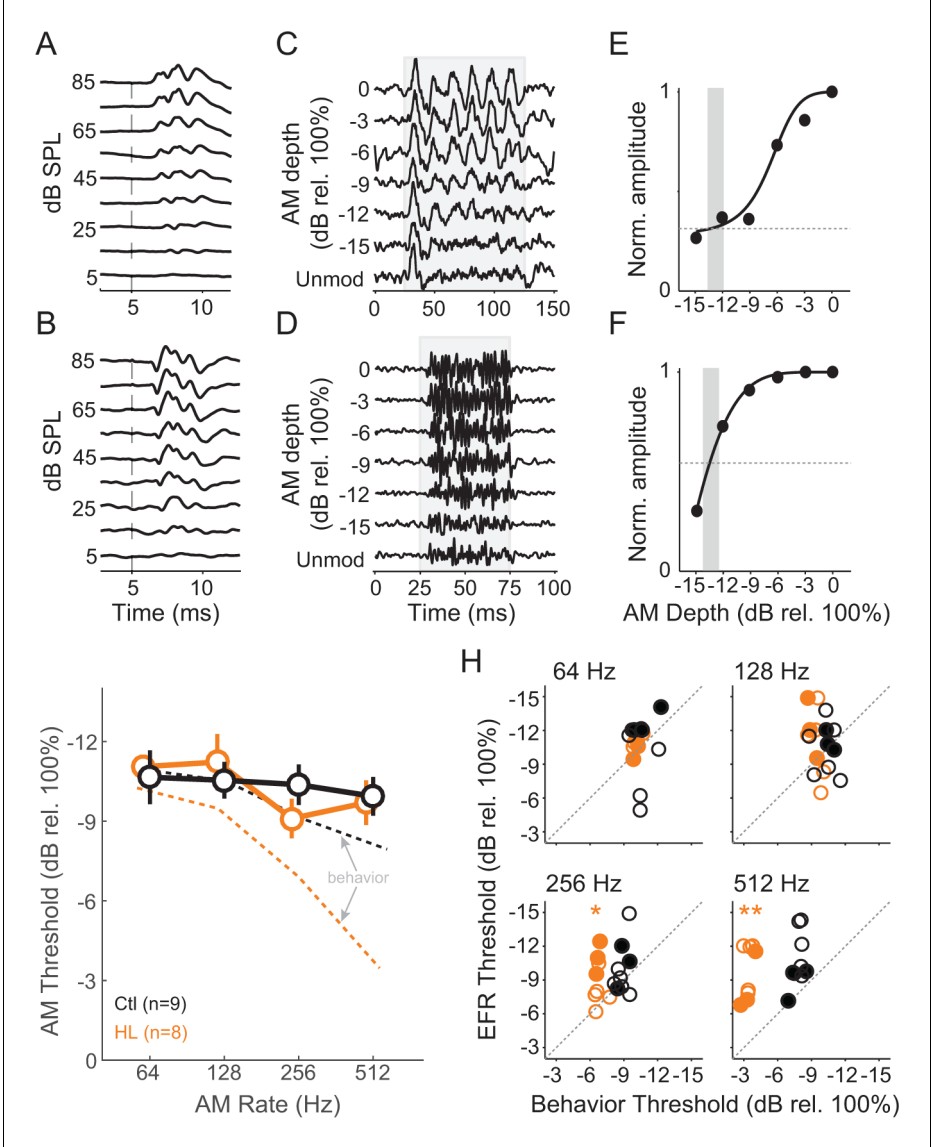

**Figure 4.** HL-related AM detection deficits cannot be explained by brainstem temporal processing. (**A,B**) Example click-evoked ABR traces across dB SPL recorded with separate electrode positions. Vertical dashed-line indicates stimulus onset. (**C,D**) Representative EFR to 64 Hz (**C**) and 512 Hz (**D**) AM noise across depths from pin electrode positions in (**A**) and (**B**), respectively. (**E,F**) Normalized FFT values as a function of AM depth corresponding to 64 Hz (**E**) and 512 Hz (**F**). (**G**) Average temporal modulation transfer functions extracted from EFRs from Ctl (black) and HL (orange) animals. Mean ± SEM. Dashed lines represented average behavior TMTFs. (**H**) Within groups post-hoc comparisons between EFR versus behavior detection thresholds across 64, 128, 256, and 512 Hz AM rates. *p<0.05; **p<0.0001; Holm-Bonferroni- corrected. Filled symbols correspond to gerbils used for awake-behaving ACx recordings.

DOI: https://doi.org/10.7554/eLife.33891.005

condition (vertical gray bars in *Figure 4E,F*). EFR thresholds were calculated from all tested AM rates, thereby deriving auditory brainstem TMTFs (*Figure 4G*). The TMTFs obtained from EFRs were not significantly different for Ctl and HL groups (two-way Mixed Model ANOVA; interaction: $F_{(3,45)}$ = 0.84, p>0.05), suggesting that our recordings do not reveal a HL-related impairment. Note that similar to psychometric assessment, stimuli were presented at equal sensation level for Ctl and HL animals (i.e. stimuli that were 40–45 dB greater for HL animals to compensate for threshold differences shown in *Figure 3A*).

Since behavioral and EFR-based AM thresholds were acquired in the same animals, we compared these measures for each group (*Figure 4H*). Ctl animals displayed similar behavior and EFR-based AM thresholds (two-way repeated measures ANOVA; interaction: $F_{(3,42)} = 1.81$, $p>0.05$). In contrast, behavior and EFR-based AM thresholds were significantly different for the HL group (two-way repeated measures ANOVA; interaction: $F_{(3,42)} = 12.6$, $p<0.0001$). Post-hoc analyses indicated behavioral thresholds were markedly worse than EFR thresholds for 256 Hz (two-tailed t-test, Holm-Bonferroni-corrected; $t = 4.09$, $p<0.05$) and 512 Hz (two-tailed t-test, Holm-Bonferroni-corrected; $t = 8.31$, $p<0.001$). These results suggest that our recorded EFR measurements do not account for the HL-induced perceptual deficits at fast AM rates. Therefore, we tested the hypothesis that HL-induced deficits in auditory cortical processing can explain perceptual performance in HL animals.

## Perceptual deficits are correlated with degraded auditory cortical sensitivity

To determine whether HL-induced perceptual deficits could be attributed to neural coding deficits downstream of the auditory brainstem, we recorded neural responses from the ACx of Ctl (n = 3) and HL (n = 3) gerbils as they performed the AM detection task. These animals also contributed to the brainstem data presented in *Figures 3* and *4* (filled symbols). Raw data (*Figure 5B*) were preprocessed to extract candidate waveforms for offline spike sorting procedures (see Materials and Methods). Principal component (PC) clustering (*Figure 5C*) was used to further sort the extracted waveforms into clusters classified as single- or multi-units. Anatomically confirmed electrode tracks within ACx are shown for one gerbil in *Figure 5D*. Example raster plots and corresponding post-stimulus time histograms (PSTHs) for one unit recorded during one session are shown in *Figure 5E* in response to unmodulated noise (Nogo signal) and fully modulated (0 dB rel. 100% depth) 256 Hz

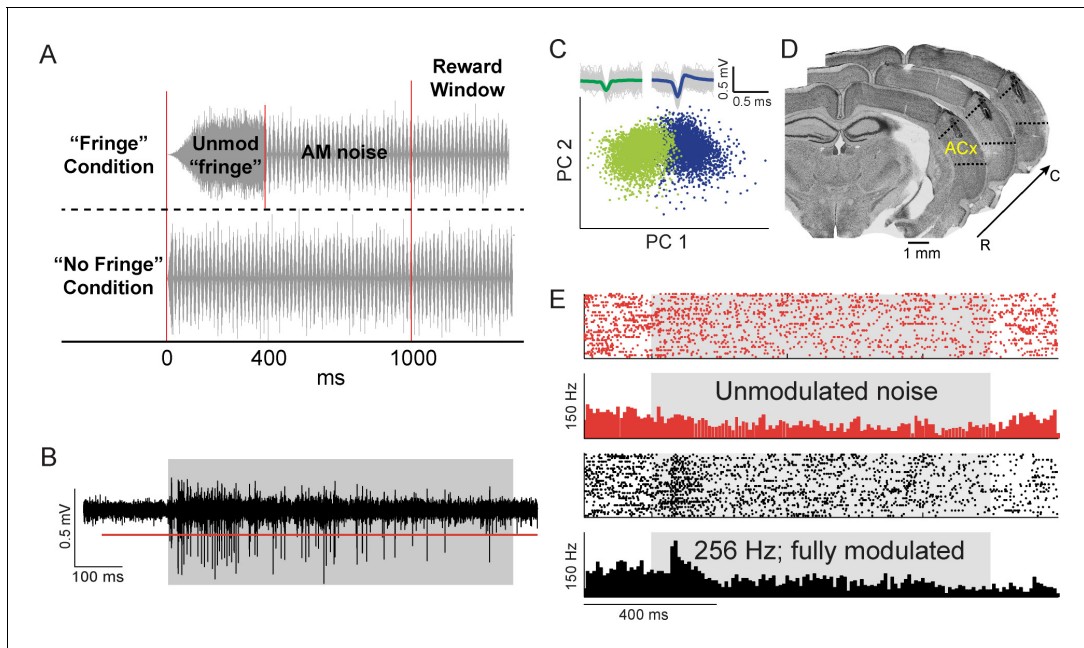

**Figure 5.** Candidate waveform selection for neurometric analyses. (A) Schematic of AM stimuli. For a majority of sessions, unmodulated and AM stimuli had a 200 ms onset ramp, followed by an unmodulated period of 200 ms that would maintain as unmodulated or transition to an AM signal ('Fringe' Condition). For the remaining number of sessions, the stimuli had cosine-ramped onsets with a 25 ms rise and fall time ('No Fringe' Condition). (B) Raw waveform trace of neural response to AM noise (gray shaded region). Red line represents selection criteria of >4 SDs above noise floor. (C) Principal component analysis plot where two waveform clusters (green and blue) are separated. Raw waveforms (gray lines) and averages (green and blue) are displayed within insets. (D) Representative electrode track from Nissl-stained coronal sections from one animal. Sections are arranged from rostral ('R') to caudal ('C') spanning 180 μm and show electrode track and lesion sites within ACx. (E) Example raster and PSTH for one unit in response to unmodulated noise (red) and 256 Hz AM at 100% modulation depth (black). Gray shaded region indicates period when stimulus was present.
DOI: https://doi.org/10.7554/eLife.33891.006

AM noise (Go signal). We recorded from a total of 460 units where 21% were classified as single-units.

ACx responses were quantified based on vector strength (VS), as well as firing rate (FR), across AM rates and depths presented during behavioral sessions. We examined the phase-locking capability of recorded ACx units by calculating VS values across AM depth for each rate tested. Consistent with a broad literature, significant phase locking was not observed for ACx units across all AM rates tested, regardless of hearing status (median VS = 0.10, interquartile range = 0.07–0.14; median Rayleigh statistic = 3.99, interquartile range = 2.87–5.66). Thus, we used FR for initial quantifications of ACx unit sensitivity to fast AM rates. Here, FR refers to a time-averaged firing rate that does not preserve temporal information, which is separate from a 'population-averaged firing rate' that may preserve temporal information.

We initially compared standard response properties of ACx units. *Figure 6A* plots cumulative distributions of ACx unit maximum stimulus-evoked ('Driven') FR, spontaneous ('Spont') FR, and coefficient of variation (CV), which represents firing-rate-normalized response variability. We found that driven and spontaneous FRs were significantly greater among Ctl units (two-sample Kolmogorov-Smirnov test; Driven FR: K = 0.12, p=0.005; Spont FR: K = 0.27, p<0.0001). CV was significantly lower among Ctl units (two-sample Kolmogorov-Smirnov test; K = 0.24, p<0.0001). Spontaneous activity tended to be fairly high, and occasionally higher than the driven FR suggesting that there was significant inhibition after stimulus onset. Similar results were obtained for the sub-population of single units. Specifically, driven and spontaneous FRs were significantly greater among Ctl units (two-sample Kolmogorov-Smirnov test; Driven FR: K = 0.32, p<0.0005; Spon FR: K = 0.39, p<0.0001). In addition, CV was significantly lower among Ctl single units (two-sample Kolomogorov-Smirnov test; K = 0.25, p<0.01).

The number of units recorded from each individual animal ranged from 80 to 285 (single units: 16–50). To account for differences in the total number of units recorded from any particular animal, we assessed group differences in FR, Spont FR, and CV with a bootstrapped statistical test (see Materials and methods). Corresponding distributions of these between group comparisons are shown in *Figure 6A* (inset). Using this more conservative approach, we found no significant difference in FR between Ctl versus HL units (confidence interval (CI) = [−3.09 2.99], p>0.05). However, Spont FR remained significantly higher among Ctl units (CI = [1.59 9.04], p<0.05) and CV remained significantly lower among Ctl units (CI = [−0.14–0.04], p<0.05). In addition, we analyzed single- and multi-unit data separately with the same bootstrapped statistical procedure and found that results were consistent across single- and multi-unit populations (*Figure 6—figure supplement 1*). Thus, whereas the difference in overall FR might be related to unit sampling, responses from ACx units in HL-reared animals were significantly less reliable.

FR responses at 256 Hz are shown for three example ACx units in *Figure 6B* (left). ACx neurons can respond to AM stimuli with either increases or decreases in FR activity (*Johnson et al., 2012*), and display greatest sensitivity at lower modulation depths (*Middlebrooks, 2008a; 2008b*). Increases in FR with greater AM depth was observed in 51.7% of recorded units (e.g., *Figure 6B*, left; open circles and triangles), whereas the remaining 48.3% of units displayed a decrease in FR with increasing AM depth (e.g., *Figure 6B*, left; open squares). Regardless of the whether the FR increased or decreased, neurometric functions were plotted for each unit as FR-based sensitivity ($d'_{FR}$) as a function of AM depth (see Materials and methods). Three example neurometric functions are shown in *Figure 6B* (right), and correspond to the FR functions plotted to the left. Note that neural sensitivity was quantified by an absolute $d'$ metric, signifying the statistical significance between FRs evoked by AM noise (Go signals) versus unmodulated noise (Nogo signal). For neurons in which FR decreased as a function of depth, $d'$ values are presented as its corresponding absolute (nonnegative) value. ACx neural threshold was defined as the AM depth at which each unit's neurometric function crossed $d'=1$. *Figure 6C* shows the distributions of FR-based AM thresholds from Ctl and HL units. The distributions did not differ from one another (Interaction, $F_{(3,210)}$ = 1.72, p>0.05; Two-way Mixed Model ANOVA), and most unit thresholds were poorer than behavioral thresholds. Therefore, we asked whether a pattern classifier could more accurately explain behavioral sensitivity. This neurometric analysis calculates performance for each unit based on the similarity of neural responses to a template (see Materials and methods). From each unit's classifier output, we plotted classifier $d'$ as a function of AM depth and extracted threshold values ($d'=1$). This analysis was conducted with four separate metrics: FR, K-means, Rcorr, and van Rossum (see

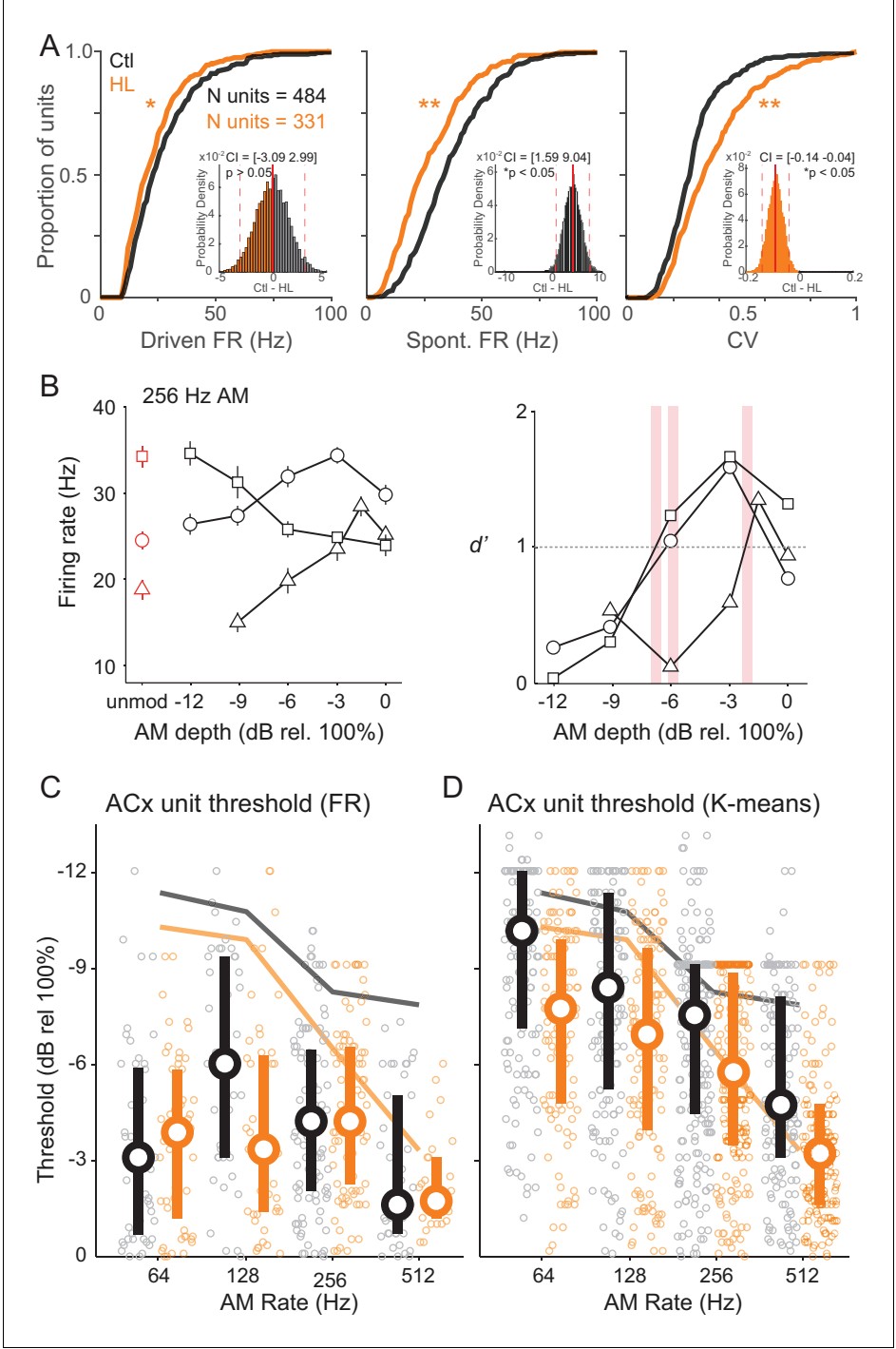

**Figure 6.** Cortical representation of fast AM rates. (**A**) Cumulative distributions of maximum stimulus-driven FR (left), spontaneous FR (middle), and CV (FR standard deviation/average FR; right) for Ctl (black) and HL (orange) units. Inset: Corresponding distributions of average between group differences from a bootstrapped statistical test. Solid and dashed red vertical lines represent the distribution average and 98.75% (two-tailed) confidence intervals (CIs), respectively. A statistically significant difference between groups tested at p<0.05 was determined if zero fell outside the 98.75% CIs. (**B**) Left: Firing rate responses from three example ACx units to 256 Hz AM noise across modulation depths (mean ± SEM). Right: Corresponding neurometric functions. Red vertical bars indicate each unit's threshold (d'=1). (**C**) Distribution of FR-based AM thresholds across AM rates for Ctl and HL units. (**D**) Distribution of classifier-based (K-means) AM thresholds across AM rates. For C and D, symbols represent

*Figure 6 continued on next page*

*Figure 6 continued*

individual units, whereas large circles and thick vertical lines represent medians and interquartile range, respectively. Thin solid lines represent average behavior TMTFs.

DOI: https://doi.org/10.7554/eLife.33891.007

The following figure supplement is available for figure 6:

**Figure supplement 1.** Multi- and single-unit data were analyzed separately with a bootstrapped statistical procedure.

DOI: https://doi.org/10.7554/eLife.33891.008

Materials and Methods). We pooled threshold values across Ctl and HL groups and compared each unit's neural classifier threshold to its corresponding psychometric performance session. The K-means metric (Pearson's r = 0.46) outperformed the FR (Pearson's r = 0.38), Rcorr (Pearson's r = 0.30), and Van Rossum (Pearson's r = 0.25) metrics, and displayed the strongest correlation with psychometric thresholds. As illustrated in *Figure 6D*, a greater percentage of units displayed K-means thresholds that were similar to psychometric performance (Ctl units: 38.9%; HL units: 38.6%). However, we did not find a significant interaction between Ctl and HL groups across AM rates ($F_{(3,768)}$ = 0.71, p>0.05; Two-way Mixed Model ANOVA). Overall, these results suggest that the unit-by-unit analyses of cortical AM encoding do not entirely reflect the HL-induced perceptual deficits. Therefore, we next asked whether AM depth was better represented at the population level.

## An ACx population code best accounts for HL-induced perceptual deficits

To examine population encoding, we constructed linear classifiers using support vector machines (SVM). Specifically, to quantify how well ACx populations could differentiate Go (AM) versus Nogo (unmodulated) signals, we used a linear population readout (*Figure 7A*), as described in Materials and methods. We trained each population using a linear readout scheme and tested the representation with ACx responses from the AM detection task. This measure estimates the ability of the ACx population to encode AM depth, and assumes that the representation is decoded by downstream neurons as a thresholded sum of weighted synaptic inputs. The parameters of our linear classifier (i. e., comparing populations of each individual Go signal versus the Nogo signal) were chosen because the animal's goal was to indicate and report the presence of AM noise. Similar to the unit-by-unit neurometric measure reported in the previous section, we examined whether the population activity from ACx units would be in accordance with the behavior differences between Ctl and HL groups (*Figure 1A and B*). Note that we included only units from which thresholds could be extracted based on FR (*Figure 6C*). The percentages of units included in this analysis were relatively equal between groups at each AM rate (64 Hz: Ctl = 49/176 (28%), HL = 53/138 (38%); 128 Hz: Ctl = 46/178 (26%), HL = 48/189 (25%); 256 Hz: Ctl = 100/270 (37%), HL = 92/247 (37%); 512 Hz: Ctl = 37/282 (13%), HL = 29/234 (12%)). We first assessed population decoding performance as a function of the number of units used in the linear population readout. For this analysis, we applied a resampling procedure to randomly select a subpopulation of units (10–100% of total units) across 250 iterations. Specifically, during each iteration of the resampling procedure, a new subpopulation of units was randomly selected (without replacement) from all units prior to the decoding readout procedure. Average performance and variability (±1 standard deviation) were calculated across 250 iterations. *Figure 7C* displays the decoding readout performance as a function of the number of units for each AM rate tested from both groups. Both Ctl and HL groups displayed greater *d'* with increasing unit counts with the highest *d'* values seen when all units within each group were utilized. Maximum decoder performance was uniformly better among units from both groups tested at the 256 Hz AM rate. This could be due to a sampling bias since we would initially search for unit responses that were driven by 256 or 512 Hz AM noise during each electrode advancement (see Materials and Methods).

To determine the sensitivity of our ACx unit population to AM rates across depth, we assessed the ability of each population to classify Go versus Nogo signals. Results from the decoder are displayed in *Figure 7D* and fitted with sigmoidal functions. Typically, higher AM depths yielded greater *d'* values for both groups, but decoder performance from HL units was worse than Ctl units at lower AM depths. For AM rates at which a clear behavioral deficit was observed for HL animals (256 and

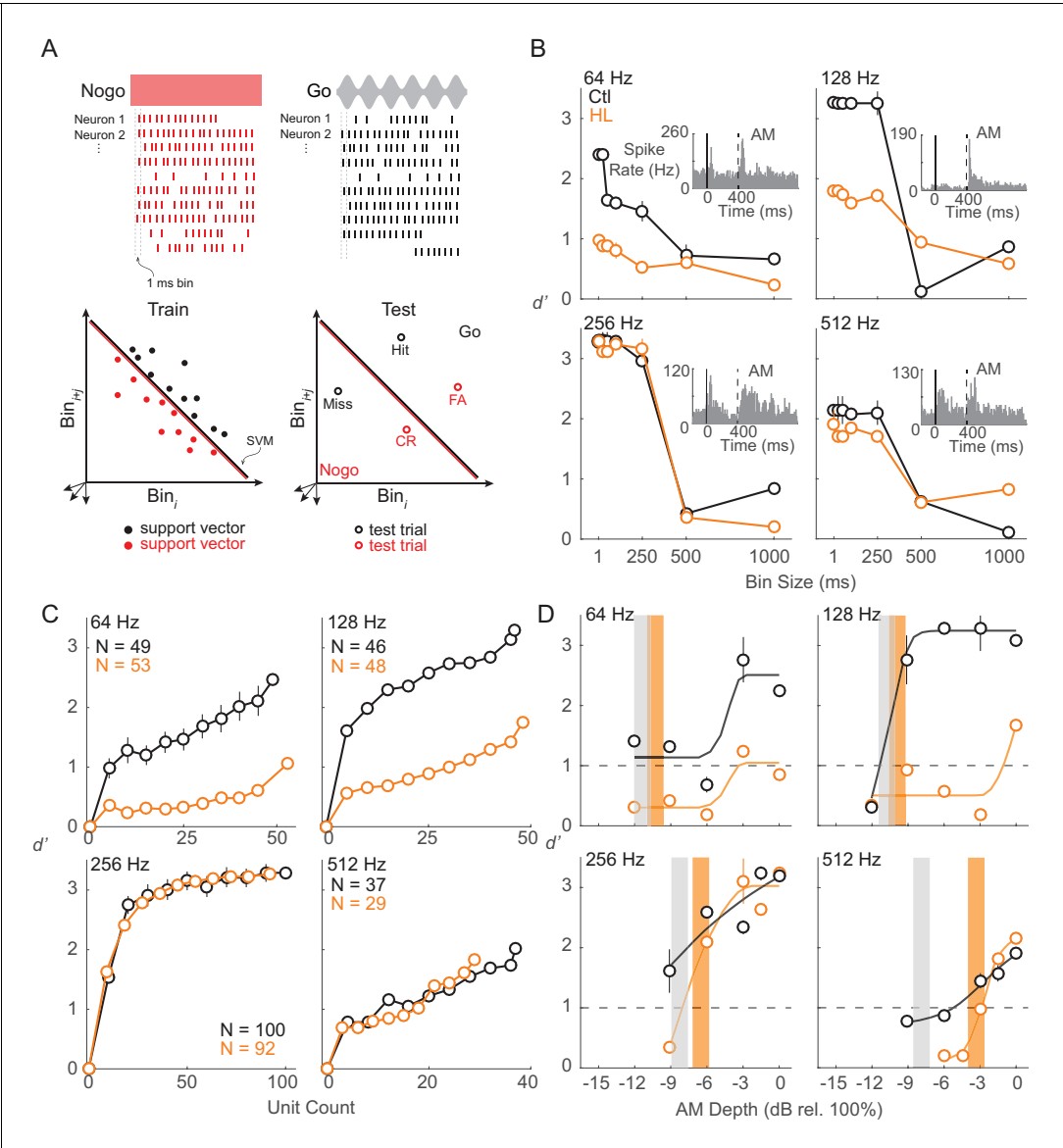

**Figure 7.** ACx population decoder analysis can explain HL-related behavioral deficits. (**A**) Assessing population encoding by measuring discriminability with a linear population readout. Hypothetical population responses for single trials of a Nogo (red) and Go (black) stimulus. Spikes were counted across 1 ms bins. Spike firing responses from *N* neurons were counted across 1 ms bins to *T* trials of *S* stimuli ('Go' and 'Nogo') and formed a population 'response vector'. 80% of trials were randomly sampled (without replacement) and averaged across *N* neurons, reducing the response vector to length N$_{bin}$ (shown across two dimensions here), and fitted to a linear hyperplane that was determined by a support vector machine (SVM) procedure ('train set'). Symbols represent 'support vectors', which are points used to create the linear boundary. Cross-validated classification performance was assessed on the remaining 20% of trials ('test' set). (**B**) Average population decoder performance between fully modulated AM noise ('Go') versus unmodulated noise ('Nogo') as a function of bin size. Example unit responses to fully modulated AM noise are shown within each subpanel. Vertical solid and dashed lines represent unmodulated noise fringe and AM noise onset, respectively. (**C**) Average population decoder performance between fully modulated AM noise versus unmodulated noise as a function of unit count. (**D**) Average population decoder performance (*d'*) for each tested AM rate as a function of AM depth from Ctl (black) and HL (orange) unit populations. Sigmoidal functions were fitted to the data for visual purposes and are not statistically validated. Vertical lines represent average behavior AM depth thresholds. Error bars represent ± 1 standard deviation across 250 iterations.

DOI: https://doi.org/10.7554/eLife.33891.009

512 Hz), HL population decoder performance was inferior at lower depths, near behavioral threshold (256 Hz: ≤ −6 dB rel. 100%; 512 Hz: ≤ −3 dB rel. 100%).

To evaluate the relative contribution of different subpopulations of units to the decoder readout, we separated units based on their monotonicity index (MI; see Materials and methods). MI distributions across each tested AM rate are shown in *Figure 8A*. The disparity between monotonically decreasing versus monotonically increasing units was smallest for 128 (monotonically decreasing versus monotonically increasing, Ctl: 45.7% versus 54.3%; HL: 31.3% versus 68.7%) and 256 (monotonically decreasing versus monotonically increasing, Ctl: 18% versus 82%; HL: 40.2% versus 59.8%) Hz conditions, whereas very few units were classified as monotonically decreasing at 64 and 512 Hz within both groups. Thus, we compared decoder readout performance between monotonically decreasing versus monotonically increasing units at 128 and 256 Hz conditions (*Figure 8B*). To account for differences in subpopulation sizes, we applied a resampling procedure to randomly select (without replacement) an equal number of units (n = 10) within each monotonic class prior to the decoding readout analysis. This procedure was conducted across 250 iterations. Decoder performance was better for monotonically decreasing units across most AM depths at 128 and 256 Hz from both Ctl and HL groups. In addition, Ctl monotonically decreasing units yielded greater *d'* values compared to HL monotonically decreasing units across all AM depths.

Overall, these results indicate that the HL-related AM detection deficits (*Figure 1*) are reflected in the population activity of ACx units. Note that our population linear readout accounted for precise temporal variability by constructing population vectors across 1 ms time bins for every trial. We reported earlier that trial-to-trial variability in FR (CV; *Figure 6A*, right) was significantly greater among HL units. It is possible that greater trial-to-trial variability among ACx responses of HL units may account for the differences seen in the population decoder outputs between Ctl versus HL units (*Figure 7D*). To examine this, we calculated a CV metric across the population vectors used in our decoder analysis. Specifically, on each trial, the population-summed spike count was calculated separately for each 1 ms bin. We then calculated the standard deviation for each trial across the bins

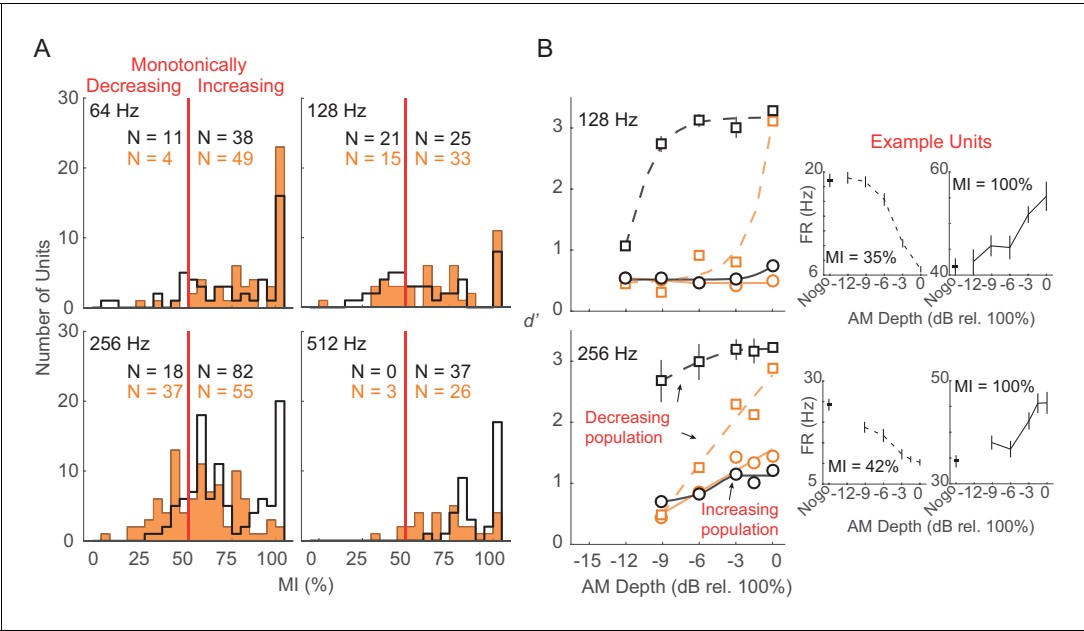

**Figure 8.** Monotonically increasing versus monotonically decreasing units. (**A**) Distribution of monotonicity index (MI) values from all Ctl (black) and HL (orange) units across 64, 128, 256, and 512 Hz AM rate conditions. Solid vertical red line represents the classification cutoff value (MI = 50%). Units with MI values ≤ 50% (>50%) were classified as monotonically decreasing (monotonically increasing). (**B**) Average population decoder performance from subpopulations of monotonically decreasing (dashed lines) and monotonically increasing (solid lines) units. Sigmoidal functions were fitted to the data for visual purposes and are not statistically validated. Data from 64 and 512 Hz conditions are not shown since very few monotonically decreasing units were classified. Error bars represent ± 1 standard deviation across 250 iterations. Individual example monotonically decreasing and monotonically increasing units are shown (B, right) with error bars representing ± 1 SEM.
DOI: https://doi.org/10.7554/eLife.33891.010

and divided by the average population-summed spike count calculated across the same bins. We found that this variability metric was significantly greater for HL versus Ctl units across all AM rates and depths tested (two-sample Wilcoxon Rank Sum test; p<0.001, z = −2.63 to −16.1).

We next asked whether HL-induced deficits in the cortical representation of fast AM reflects degraded choice probability (CP). CP quantifies trial-to-trial correlation between FR and an animal's behavioral choice via an ROC-based ideal observer (see Materials and Methods). We calculated CP across two separate time windows, 0–400 and 400–800 ms relative to AM onset (*Figure 5A*). These time windows were selected because most of the stimulus-driven information was found at 0–400 ms, whereas animals made physical choices at 400–800 ms. Regardless of hearing status group, CP values did not differ from chance, either during the 0–400 ms time window (H$_0$: average CP value = 0.50; Ctl: p>0.05, t = −0.03; HL: p>0.05, t = 1.75; one-sample t-test), or the 400–800 ms window (Ctl: p>0.05, t = 0.24; HL: p>0.05, t = 1.52; one-sample t-test). Overall, these findings suggest that our recorded ACx units did not represent the animals' behavioral choice during task performance.

## Discussion

Sensory deprivation during development can induce detrimental changes to CNS function, and may result in perceptual impairments. While many normative studies have utilized awake-behaving preparations, none have exploited the paradigm to simultaneously assess neural and perceptual plasticity following early sensory deprivation (*Knudsen et al., 1982*; *1984a*; *1984b*; *Mogdans and Knudsen, 1993*; *1994*; *Raggio and Schreiner, 1999*; *Snyder et al., 2000*; *DeBello et al., 2001*; *Moore et al., 2002*; *Yu et al., 2005*; *Takahashi et al., 2006*; *Fallon et al., 2008*; *Razak et al., 2008*; *Popescu and Polley, 2010*; *Rosen et al., 2012*; *Polley et al., 2013*). Furthermore, neural studies suggest that, depending on the type of deprivation and its age of onset, many subcortical structures can display functional impairments (for review, see *Sanes and Woolley, 2011*; *Sanes, 2013*). Therefore, the extent to which neural processing deficits in primary sensory cortex are associated with diminished perceptual performance is uncertain.

Here, we report that developmental HL impairs behavioral detection of temporal envelope cues (*Figure 1B*) that support aural communication and the perceptual qualities of rhythm, prosody, periodicity pitch, and musical attack. These findings are consistent with clinical studies that report poorer AM sensitivity at fast rates (>100 Hz) in profoundly deaf children, congenitally deaf adults, and even late onset HL adults (*Formby, 1987*; *Grant et al., 1998*; *Park et al., 2015*). Since basic performance metrics did not differ significantly between groups (*Figure 1E–G*), the perceptual impairments point to a sensory processing deficit. A measurement of temporal processing at the level of the auditory brainstem, the envelope following response (EFR), was not affected by developmental conductive HL. In contrast, degraded ACx population-level encoding was sufficient to explain HL-associated behavioral deficits (*Figure 9*).

### Fast temporal processing in normal hearing animals

The characterization of fast AM encoding in ACx is well documented across many species under conditions of anesthesia (e.g., squirrel monkey: *Bieser and Müller-Preuss, 1996*; cat: *Schreiner and Urbas, 1988*; guinea pig: *Creutzfeldt et al., 1980*; gerbil: *Schulze and Langner, 1997*). The general consensus is that very few ACx neurons synchronize their spike firing to very fast modulations (>100 Hz), and those units that encode fast rates do so with non-phase-locked responses (*Joris et al., 2004*). This is further supported by studies that have shown that whereas auditory brainstem neurons can display a synchronized response to fast modulations, such temporal responses are transformed into a firing rate code in downstream regions, including ACx (*Palmer, 1982*; *Schreiner and Urbas, 1988*; *Frisina et al., 1990*; *Preuss and Müller-Preuss, 1990*; *Joris and Yin, 1992*; *Müller-Preuss et al., 1994*; *Rhode and Greenberg, 1994*; *Gaese and Ostwald, 1995*; *Bieser and Müller-Preuss, 1996*; *Krishna and Semple, 2000*; *Lu and Wang, 2000*; *Lu et al., 2001*; *Liang et al., 2002*; *Joris et al., 2004*; *Bendor and Wang, 2007*; *Wang et al., 2008*; *Bendor and Wang, 2010*). Thus, a cortical rate code is thought to sufficiently account for AM perceptual sensitivity (*Niwa et al., 2012*; *2013*; *2015*; *von Trapp et al., 2016*; *Caras and Sanes, 2017*) with improved sensitivity corresponding with increasing AM depth (*Bieser and Müller-Preuss, 1996*; *Liang et al., 2002*; *Johnson et al., 2012*; *Rosen et al., 2012*). Our current data is consistent with this literature. Maximum phase-

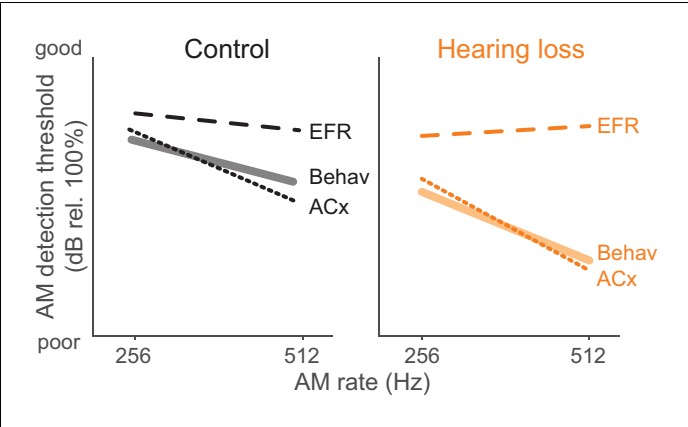

**Figure 9.** Summary of psychometric, EFR, and cortical threshold values. Solid lines represent behavior thresholds. Dashed lines represent EFR thresholds. Dotted lines represent ACx population decoder thresholds. Ctl ACx population threshold at 256 Hz was taken as the lowest AM depth presented since decoder output *d'* values did not extend below 1. Only 256 and 512 Hz conditions are shown to summarize our main findings that population ACx activity contains sufficient sensory information to explain behavioral detection of very fast AM rates and can account for the perceptual deficits displayed by HL-reared animals. The y-axis (dB rel. 100%) is the same from each of the primary data graphs from which this schematic emerges (Psychophysics: *Figure 1*; EFR: *Figure 4*; ACx: *Figure 7*).
DOI: https://doi.org/10.7554/eLife.33891.011

locking capability among our ACx units in response to AM rates of 64–512 Hz was very low, and generally not significant. In contrast, FR carried significant information about AM depth. Regardless of the minimal phase-locking of our ACx units to fast AM rates, our population data indicate that decoding performance increases as we decrease bin size (*Figure 7B*), suggesting that the precise temporal structure of the population-averaged response is an important factor to representing fast AM rates.

A neural population can represent vast amounts of sensory information, but may not all be decoded during perception, which would lead to a discrepancy between the output of a decoding readout versus behavioral performance (*Figure 7D*). Alternatively, common characteristics of the unit population may inflate the decoding readouts relative to psychophysical sensitivity. For example, we found that a linear classifier readout procedure that consisted of only monotonically decreasing units outperformed one comprised of only monotonically increasing units (*Figure 8B*). If the optimal decoding of fast AM information occurs among monotonically decreasing units, then this could explain the superior *d'* values observed for this class of neurons at lower AM depths (*Figure 8B*). That said, the inclusion of all neurons more closely resembles psychometric performance as previously suggested (*Shadlen et al., 1996*). Overall, our findings in awake-behaving animals indicate that population ACx activity contains sufficient sensory information to explain behavioral detection at very fast AM rates and can account for the perceptual deficits displayed by HL-reared animals.

## Neural basis of hearing loss-induced perceptual deficit in fast temporal processing

The primary goal of this study was to compare both brainstem and cortical processing to behavioral measures of AM detection. Thus, we obtained AM depth thresholds from brainstem EFRs, which have been used to assess brainstem temporal processing across a broad range of AM rates in humans and non-human species (*Boettcher et al., 2001*; *2002*; *Leigh-Paffenroth and Fowler, 2006*; *Parthasarathy et al., 2010*; *2014*; *2016*; *Parthasarathy and Bartlett, 2011*; *2012*). We found that EFR sensitivity did not differ significantly between HL and Ctl animals (*Figure 4G*). In fact, EFR thresholds from HL animals were better than psychometric performance at very fast AM rates (*Figure 4H* and *Figure 8*, right). Since EFRs were not simultaneously measured while animals were performing the task, it is possible that similar measurements under awake-behaving conditions may

reveal interesting changes. For example, *Slee and David, 2015* have shown that inferior colliculus neurons are modulated by task-engagement, which leaves open the possibility that EFRs obtained during performance of the fast AM detection task could have been differentially modulated in Ctl and HL animals. Another possibility is that a subpopulation of brainstem neurons is impaired following developmental HL, but this impairment is masked with a population level assessment function, such as our ABRs and EFRs. Regardless, our EFR measurements do not reveal a HL-related impairment that could account for behavioral deficits in fast AM rate detection. In this light, our findings point towards degraded neural encoding beyond the auditory brainstem as the neural origin of perceptual deficits following early auditory deprivation.

Whole-cell recordings from ACx brain slices have revealed that developmental conductive HL disrupts ACx synaptic and membrane properties (*Xu et al., 2007*; *Takesian et al., 2010*; *2012*), even for temporary manipulations (*Mowery et al., 2015*; *2016*). Therefore, it is plausible that mechanisms intrinsic to gerbil ACx could account, in part, for HL-induced behavioral deficits (*Rosen et al., 2012*; *Buran et al., 2014a*; *Caras and Sanes, 2015*; *Ihlefeld et al., 2016*; *von Trapp et al., 2016*). Consistent with this conceptual framework, we found that maximum stimulus-driven and spontaneous firing rates were significantly weaker among ACx units recorded from HL versus Ctl animals (*Figure 6A*). Furthermore, trial-to-trial response variability (CV) was greater among ACx units in HL-reared animals. Since ACx synaptic excitation and inhibition are normally in approximate balance (*Tan et al., 2004*; *2007*; *Wu et al., 2008*; *Tan and Wehr, 2009*; *Zhou et al., 2014*), a net reduction of inhibition could diminish stimulus selectivity. This so called 'iceberg effect' implies that inhibition plays a crucial role in accurate sensory-evoked activity (*Rose and Blakemore, 1974*; *Carandini and Ferster, 2000*; *Isaacson and Scanziani, 2011*). Thus, the degraded synaptic inhibition observed following developmental HL may explain, in part, some of the observed behavioral impairments.

Since behavioral performance in HL animals was not explained by brainstem EFR responses (*Figure 4G*) or ACx unit sensitivity (*Figure 6C–D*), we asked whether ACx population activity was degraded by HL. In fact, we found a robust correlate between HL-related behavioral deficits and population-level activity in ACx (*Figure 7*). However, our observed deficits in HL-related cortical population responses may not necessarily produce behavioral deficits since AM detection thresholds at slower modulation rates of 64 Hz were similar between Ctl versus HL animals (see *Figure 1B*). It might be the case that higher levels of the auditory system are able to remediate the deficit for slower modulation rates, but not for faster modulation rates. Future studies that compare ACx versus downstream processing in animals with developmental HL are needed to better assess this issue. Regardless, the finding of superior population-level representation of task-related performance compared to individual unit metrics is consistent with previous studies from visual (*Pouget et al., 2000*; *Averbeck et al., 2006*; *Rust and Dicarlo, 2010*; *Graf et al., 2011*; *DiCarlo et al., 2012*; *Pagan et al., 2013*; *Pagan and Rust, 2014*), somatosensory (*Safaai et al., 2013*; *Adibi et al., 2014*), and auditory cortices (*Middlebrooks et al., 1994*; *1998*; *Xu et al., 1998*; *Miller and Recanzone, 2009*; *Ince et al., 2013*; *Pachitariu et al., 2015*; *Christison-Lagay et al., 2017*). The current results provide an experimental test of these ideas by identifying a cortical population-level contribution to sensory deprived-related perceptual deficits. Developmental plasticity induced by abnormal changes to acoustic signals can also lead to improved sensory encoding and perception than what would have otherwise occurred (*Keating et al., 2015*). It is possible that, with additional training, our animals would have eventually learned to perform the AM detection task at control levels.

Our population analysis utilized a classifier that was based on a linear population readout (support vector machine procedure) that took into account neuronal variability across 1 ms time windows. We found that ACx population activity from animals reared with conductive HL were more variable across trials compared to the unit population from Ctl animals. This signifies the relationship between higher trial-to-trial response variability and impaired psychometric performance in HL animals. Overall, our current results support the notion that psychometric performance on an AM detection task benefits from lower response variability (*von Trapp et al., 2016*) since HL animals performed worse than Ctl animals and possessed less reliable cortical responses. We conclude that developmental HL detrimentally increased CV of ACx population activity (*Figure 6A*), possibly by reducing cortical inhibition. Our results are consistent with the interpretation that early auditory deprivation diminishes the neural information that could be used to represent fast AM signals. The HL-induced neural encoding deficits are not observed at the brainstem, but reflect changes at higher stages in the auditory CNS network, such as ACx population activity.

## Materials and methods

### Subjects

Mongolian gerbils (*Meriones unguiculatus*, N = 20, 12 male) were weaned from commercial breeding pairs (Charles River) and housed on a 12 hr light/12 hr dark cycle. All procedures were approved by the Institutional Animal Care and Use Committee at New York University. No a priori analysis regarding sample size was performed during study design. Having previously examined HL-induced perceptual deficits with both aversive and appetitive procedures, we found that variance is relatively smaller for the latter (*Rosen et al., 2012*; *Buran et al., 2014a*; *Sarro et al., 2015*; *Caras and Sanes, 2015*; *Ihlefeld et al., 2016*; *von Trapp et al., 2016*). Thus, 5–8 animals can be used to identify behavioral deficits with the appetitive procedure used here. Overall, each experiment was performed once with technical replication occurring for behavioral data only (i.e., each animal was tested psychometrically multiple times), and all measures were subject to biological replication.

### Hearing loss surgery

For the conductive hearing loss (HL) group (n = 8, five male), bilateral conductive hearing loss was induced at postnatal day 10, just before ear canal opening as described previously (*Xu et al., 2007*). A surgical level of anesthesia was induced, the tympanic membrane exposed and punctured, and the malleus removed with forceps. The ear canals were left intact. The gerbils were returned to their home cages for recovery and reared to adulthood with a permanent bilateral conductive loss of approximately ~40 dB (*Figure 3A*; *Tucci et al., 1999*; *Buran et al., 2014a*; *Rosen et al., 2012*).

### Psychophysical testing

#### Behavioral apparatus

Adult gerbils were placed in a plastic test cage (0.25 × 0.25 × 0.4 m) in a sound-attenuating booth (Industrial Acoustics; internal dimensions: 2.2 × 2 × 2 m) and observed via a closed-circuit monitor. Acoustic stimuli were delivered from a calibrated free-field tweeter (DX25TG0504; Vifa) positioned 1 m above the test cage. Sound calibration measurements were made with a 1/4-inch free-field condenser recording microphone (Brüel and Kjaer) placed in the center of the cage. Stimulus, water reward delivery, and behavioral data acquisition were controlled by a personal computer through custom MATLAB scripts (written by Dr. Daniel Stolzberg: https://github.com/dstolz/epsych) and an RZ6 multifunction processor (Tucker-Davis Technologies).

#### Training

Perceptual sensitivity was assessed with a positive reinforcement Go-Nogo appetitive conditioning paradigm, as described previously (*Buran et al., 2014a*; *2014b*; *Sarro et al., 2015*; *von Trapp et al., 2016*; *Ihlefeld et al., 2016*; *von Trapp et al., 2017*). Briefly, gerbils were placed on controlled water access, and trained to detect amplitude modulated (AM) frozen broadband noise (25 dB roll-off at 3.5 kHz and 20 kHz) across modulation rates of 64, 128, 256, and 512 Hz at a modulation depth of 0 dB rel. to 100% depth. For normal hearing control (Ctl) animals, sound level was constant (45 or 50 dB equivalent SPL) for all AM stimuli. For HL animals, identical stimuli were used but presented at 90 dB SPL (i.e., 45 dB louder than that used for Ctl animals), to compensate for the induced loss. The gain of the AM noise signals were adjusted to accommodate for changes in average power across modulation depth (*Viemeister, 1979*; *Wakefield and Viemeister, 1990*). Animals were initially trained to initiate a trial by placing their noses in a cylindrical port that interrupted an infrared beam, and to approach a water lick spout upon presentation of an AM stimulus (the 'Go' signal). Water reward (25 µl) was delivered via a syringe pump (NE-1000; New Era). After learning to consistently initiate Go trials, animals were then trained to repoke upon a presentation of an unmodulated noise signal (the 'Nogo' signal). Unmodulated Nogo trials (30% probability) were randomly interleaved with Go trials.

#### Testing

Behavioral sensitivity for each tested AM rate was assessed by presenting Go trials across five different AM depths that bracketed an animal's psychometric threshold. Depths are presented here on a dB scale (rel. 100% depth). Specifically, 0 dB refers to fully modulated (100% depth) noise, whereas

negative dB values refer to shallower depths (e.g., −3, −6, −9 dB correspond to 71, 50, and 35%, respectively). For the majority of sessions, unmodulated and AM stimuli had a 200 ms onset ramp, followed by an unmodulated period of 200 ms, and then transitioned to an AM or unmodulated signal ('fringe' condition; *Figure 5A*). For the remaining number of sessions, the stimuli had cosine-ramped onsets with a 25 ms rise and fall time. A two-way ANOVA revealed detection thresholds across AM rates between fringe and no fringe stimuli were similar (Ctl: $F_{(1,3)}$ = 1.32, p>0.05; HL: $F_{(1,3)}$ = 1.71, p>0.05) and were thus grouped together. For a subset of Ctl animals (n = 4) trained on the AM detection task, we roved average sound level for Go and Nogo signals across a 12 dB SPL span (45–57 dB SPL) at 0 dB depth to assess the potential use of an average level cue, even though the gain of the AM signal was adjusted to control for changes in average power due to changes in modulation depth (*Viemeister, 1979*; *Wakefield and Viemeister, 1990*).

## Psychometric analysis

Behavioral analyses were performed on sessions where false alarm rates were ≤30%, and the animal performed a minimum of 20 trials per Go signal (total of ≥100 Go trials). The number of sessions that fit this criterion ranged from 3 to 15 per tested AM rate for each animal. Responses were scored as a Hit (False Alarm) when gerbils approached the water spout on a Go (Nogo) trial. The percentage of Hits were plotted as a function of modulation depth and these psychometric functions were fit with a cumulative Gaussian using Bayesian inference from the open-source package psignifit 4 for MATLAB (*Schütt et al., 2016*; *Caras and Sanes, 2017*). The default priors in psignifit 4 were used since they worked well for fitting the data. The fitted distribution of percent correct scores were then transformed to the signal detection metric, *d'*, by calculating the difference in z-scores of hit rate versus false alarm rate (*Green and Swets, 1966*). We constrained hit and false alarm rates to floor (0.05) and ceiling (0.95) values to avoid *d'* values that approach infinity. Psychometric functions of *d'* are plotted as a function of AM depth for each tested AM rate. Psychometric threshold was defined as the AM depth at which *d'*=1. We also measured Lapse Rate, or the probability of a Miss on the easiest Go signals (i.e., high modulation depths). Lapse rate has been used as proxy for task engagement and motivation as unmotivated animals would tend to miss easy AM depths.

The thresholds from each animal across the tested AM rates were used to construct temporal modulation transfer functions (TMTFs; [*Viemeister, 1979*]). The overall height of a TMTF (i.e., gain) characterizes the temporal encoding efficiency (*Buss et al., 2012*; *Park et al., 2015*), whereas TMTF shape (i.e., slope) indicates the cutoff limit at which temporal information can be resolved, or accurately encoded (*Hall and Grose, 1994*; *Park et al., 2015*). To quantify the gain and slope of the TMTFs, each animal's TMTF was fitted to an exponential function (*Park et al., 2015*):

$$y(m) = A \times \exp^{b \times m} \tag{1}$$

where *m* is the modulation rate and *y* is the value of the modulation detection threshold. Parameters *A* and *b* are the fitted coefficients where *b* determines the slope of the function and *A* characterizes overall TMTF gain.

## Electrophysiology

Electrophysiological procedures are identical to those of previous studies from our laboratory (*Buran et al., 2014b*; *von Trapp et al., 2016*; *Caras and Sanes, 2017*). Below, we provide a summary of the procedures.

## Electrode implantation

A subset of gerbils (n = 6; Ctl n = 3, HL n = 3) underwent electrode implantation surgery. After an initial set of psychometric functions was obtained across the four tested AM rates, the animal was anesthetized with isoflurane/O₂, secured on a stereotaxic device (Kopf), and a 16-channel silicone probe array (four shanks with recording sites arranged in a 600 × 600 µm grid; Neuronexus A4 × 4–4 mm-200-200-1250-H16_21 mm) was implanted in the left core auditory cortex. The array was fixed to a custom-made microdrive to allow for subsequent advancement across recording sessions, and angled at 25-degrees in the mediolateral plane. Typically, we aimed the rostral-most shank of the array to be positioned at 3.9 mm rostral and 4.6–4.8 mm lateral to lambda. Notably, a ground wire was inserted in the contralateral cortical hemisphere. Animals recovered for at least 1 week before

being placed on controlled water access for further psychometric testing. At the termination of each experiment, animals were deeply anesthetized with sodium pentobarbital (150 mg/kg) and electrolytic lesions were made through one contact site via passing current (7 mA, 5–10 s). Animals were then perfused with phosphate-buffered saline and 4% paraformaldehyde. Brains were extracted, post-fixed, sectioned on a vibratome (Leica), and stained for Nissl. Brightfield images were acquired with a high-resolution slide scanner (Olympus) and electrode tracks were reconstructed offline and compared to a gerbil brain atlas (*Radtke-Schuller et al., 2016*) to verify targeted core auditory cortex recordings (*Figure 5C*).

## Data acquisition

Extracellular recordings were acquired telemetrically using a wireless headstage and receiver (W16, Triangle Biosystems Inc.) while animals simultaneously performed the AM detection task. The analog signals were preamplified and digitized at a 24.414 kHz sampling rate (TB32; Tucker-Davis Technologies). The converted digital signals were then fed via fiber optic link to the RZ5 base station (TDT, Tucker-Davis Technologies) for filtering and processing. All recording channels but one were averaged together and subtracted from the remaining channel as a common average referencing technique (*Ludwig et al., 2009*). Offline, signals were first high-pass filtered (300 Hz). For each individual channel, a representative 16 s recording segment was used to calculate the standard deviation (SD) of background noise ('noise floor') using the algorithm described by (*Quiroga et al., 2004*). A spike extraction threshold was set to 4 SDs > noise floor, and an artifact rejection threshold was set to 20 SDs > noise floor. Candidate waveforms were then peak-aligned, hierarchically clustered, and sorted in principal component (PC) space using the MATLAB-based package UltraMegaSort 2000 (*Fee et al., 1996*; *Hill et al., 2011*). Well-isolated single-units demonstrated a clear separation in PC space, and fewer than 10% of refractory period violations. For the majority of recording sites that did not meet this criterion and contained spikes from several unresolved units, they were considered multi-units. Separate analyses of single- versus multi-unit populations revealed no systematic differences. Thus, we pooled the populations for all group analyses reported.

To examine cortical sensitivity to fast AM rates, we restricted our recordings to units that were driven by 256 and/or 512 Hz AM noise. To search for driven activity, we initially tested implanted animals on 256 or 512 Hz AM noise detection (Go signal: 0 dB rel. 100%) upon every electrode advancement (~80–100 μm). If we judged the electrode channels to be 'driven' by the AM stimuli (i. e., change in activity during the stimulus delivery period relative to ongoing spontaneous activity via online visual inspection of the waveform traces to each channel), the animal would proceed with psychometric testing across AM depths. On subsequent sessions, if similar channels were judged to be driven by AM stimuli, we proceeded with psychometric testing at 64 and/or 128 Hz. Neurophysiological analyses were performed on sessions that met the same criteria for conducting psychometric analyses. We found no main effect of pre- versus post-electrode implantation psychometric thresholds (Ctl: two-way mixed model ANOVA; $F_{(1,58)} = 3.09$, $p>0.05$; HL: two-way mixed model ANOVA, $F_{(1,50)} = 3.65$, $p>0.05$), arguing against the possibility of testing-order effects.

## Neurometric analysis

Spontaneous firing rate (Hz) was calculated across 200 ms prior to nose poke. This time period was chosen because animals were trained to place their noses in the nose port for 200 ms to initiate a trial. Thus, we chose a time period immediately prior to that action to avoid potential artifacts that might have been evoked with nose port contact.

Each unit's driven firing rate (FR) was calculated across a time period of spike trains corresponding to the initial onset of the modulation of the stimulus. A neurometric firing rate-based *d′* was calculated at each Go value (AM depth) by normalizing the firing rate by a SD pooled across all the stimuli (z-score), and subtracting the Nogo value (unmodulated signal) from each Go stimulus. Thus, $d′_{FR} = z(\text{AM Depth}_{Go}) - z(\text{Unmodulated}_{Nogo})$. In order to optimize our analyses under such conditions of processing across fast temporal rates, we assumed optimal detection success if the finer temporal structure of the neural responses were taken into account (*Rossi-Pool et al., 2016*). Thus, we computed mean FR for each unit across time windows of different lengths (from 25 to 1000 ms, in steps of 25 ms) for each tested AM rate and depth. The 'best time window' was indicated as the time window corresponding to the maximum *d′* calculated between Go AM depths versus the unmodulated

noise Nogo signal. The corresponding FR and *d'* values from this time window across AM depths were used to fill out the neurometric function.

The dependence of FR on modulation depth was quantified by computing a 'monotonicity index' (MI; *de la Rocha et al., 2008*; *Sadagopan and Wang, 2010*; *Middlebrooks, 2008a*; *2008b*; *Watkins and Barbour, 2011*). Specifically, the MI is the FR at 100% modulation depth expressed as a percentage of the maximum FR observed across all depths. Indices of 100% indicate units display a monotonic increase in FR with increasing modulation depth, with smaller indices indicating monotonic depth dependence with FR that declined at the maximum modulation depth. Thus, units with MI values > 50% were classified as 'monotonically increasing,' whereas units with MI values ≤ 50% were classified as 'monotonically decreasing'.

The strength of stimulus synchrony for each unit across tested AM rates and depths was represented by vector strength (VS; *Goldberg and Brown, 1969*). The VS could range from 0 (no synchrony) to 1 (all spikes at identical phase). The statistical significance of the VS was evaluated by the Rayleigh test of uniformity (*Mardia, 1972*) at the level of $p < 0.001$.

## Neurometric classifiers

In addition to the standard measure of FR, we adopted a pattern classifier analysis (*Machens et al., 2003*; *Narayan et al., 2006*; *Billimoria et al., 2008*; *Wang et al., 2007*; *Schneider and Woolley, 2010*; *von Trapp et al., 2016*) to further assess neural encoding of fast AM rates. Given that there are many potential ways cortical units can encode AM, we incorporated four neurometrics for our classifier analyses. The FR metric used the overall spike firing (spikes/sec) in response to each stimulus to calculate neural discriminability. The van Rossum (VR) metric utilizes Euclidean distance to quantify the dissimilarity between two spike trains in high-dimensional space (*van Rossum, 2001*). Spike trains that are similar to one another have smaller distances between them than spike trains that are dissimilar. The K-means metric, much like the VR metric, utilizes Euclidean distance as a measure of spike train similarity. However, the K-means utilizes an iterative clustering algorithm that classifies spike trains into *K* clusters based on their proximity to each other in high-dimensional space (*Duda et al., 2001*; *Schneider and Woolley, 2010*). The Rcorr metric used an established correlation-based measure of spike timing reliability (*Schreiber et al., 2003*; *Wang et al., 2007*; *Caras et al., 2015*). Rcorr is a normalized measure with a value that can range between 0 (no relationship) to 1 (perfect correlation) between compared spike trains. Spike trains were convolved with an exponential function with a decay constant (tau) across 2–512 ms that optimized the discriminability of each unit.

The pattern classifier was implemented as follows: First, for each individual unit, two separate templates were formed by randomly selecting (without replacement) and averaging 50% of Go and Nogo trials of spike trains. Subsequent Go and Nogo trials from this unit were selected at random without replacement and were compared with each template and classified on the basis of the smallest difference between the neurometric values described above. Go trials were labeled as 'Hits' or 'Misses' if they were classified as Go or Nogo trials, respectively. Likewise, Nogo trials were labeled as 'Correct Rejections' or 'False Alarms' if they were classified as Nogo or Go trials. This procedure was repeated 1,000 times with different Go and Nogo templates on each iteration. Neural sensitivity is quantified with a *d'* metric by incorporating overall hit and false alarm scores, as described in the psychometric analysis above.

## Neurometric thresholds

Neurometric thresholds for individual units were defined as the lowest modulation depth that proved significantly different from the unmodulated noise Nogo signal at a tested AM rate. Thus, to calculate neurometric thresholds, values of FR-based and classifier-based *d'* were plotted as a function of AM depth such that each unit was compared with its own baseline (i.e., unmodulated noise Nogo). Significant modulation depths possessed neural $d' \geq 1$. The criteria for calculating a neurometric threshold consisted of units possessing at least one significant modulation depth. This is similar to previous studies that examine representations of AM in the cortex (*Rosen et al., 2010*, *2012*) and inferior colliculus (*Nelson and Carney, 2007*). Two *d'* values and their corresponding AM depths were taken from the function: greatest *d'* value below 1, and smallest *d'* value above 1. Threshold was taken as the corresponding interpolated AM depth that crossed $d' = 1$.

## Population coding

We used a linear classifier readout procedure (*Hung et al., 2005*; *Rust and Dicarlo, 2010*; *Rust and DiCarlo, 2012*; *Pagan et al., 2013*; *Carruthers et al., 2015*; *Pachitariu et al., 2015*; *Christison-Lagay et al., 2017*) to assess AM sensitivity across a population of ACx units. Specifically, a linear classifier was trained to decode responses from a proportion of trials to each stimulus set (e.g., 'Go' and 'Nogo'; *Figure 7A*). Spike firing responses from $N$ neurons were counted across 1 ms bins to $T$ trials of $S$ stimuli ('Go' and 'Nogo') and formed a population 'response vector'. The number of trials was equal for each hearing status group. 80% of trials were randomly sampled (without replacement) and averaged across $N$ neurons, reducing the response vector to length $N_{bin}$, and fitted to a linear hyperplane that was determined by a support vector machine (SVM) procedure ('training set'). Cross-validated classification performance was assessed on the remaining 20% of trials by computing the number of times the test set was correctly classified and misclassified based on the linear hyperplane (*Figure 7A*, bottom). Performance metrics included the proportion of correctly classified Go trials ('Hits') and misclassified Nogo trials ('False Alarms'). Similar to the psychophysics and pattern classifier analyses, we converted population decoder performance metrics into $d'$ values (*Figure 7B,C and D*). We explored how well our ACx population could differentiate between Go (100% AM Depth) versus Nogo (unmodulated) signals across different bin sizes (e.g., 1–1000 ms) and found the best performance was seen with 1 ms bins (*Figure 7B*). In order to construct a homogenous neural population, absolute z-score values were calculated from the responses across each trial. This procedure was conducted across 250 iterations with a new randomly drawn train and test set for each iteration. Error bars were calculated as the SD of performance across all 250 iterations. The SVM procedure was implemented in MATLAB using the 'svmtrain' and 'svmclassify' functions with a linear kernel and cost set to 1 (default value). The C ('cost') parameter controls the decision boundary of classifying training points correctly. For a high C value, the SVM optimization will utilize a smaller-margin hyperplane, whereas a low C value will cause the SVM optimization to utilize a larger-margin hyperplane. In general, exceedingly high and low C values could misclassify many training points. We set the cost parameter to the algorithm's default value of 1, which is ideal for our train and test configuration of two classes (Go versus Nogo).

## Choice probability

To assess the relationship between neural sensitivity in ACx with behavioral choice, we calculated choice probability (CP) (*Britten et al., 1996*; *Tsunada et al., 2016*; *Christison-Lagay et al., 2017*). CP quantifies trial-to-trial correlation between neural activity and an animal's behavioral choice via an ROC-based ideal observer (*Green and Swets, 1966*). Specifically, CP is the degree of overlap between two distributions of spike firing for trials where the animals report modulation ('Go') versus trials where the animal does not report modulation ('Nogo'). Since animals rarely failed to report modulation at high modulation depths (~0 dB rel. 100% depth), we only included modulation depths around threshold where a substantial number of reported modulation and no modulation trials were made. A CP value of 1 (or 0) indicates firing rate is always higher (or lower) when the animal reports an AM stimulus than when the animal does not. A CP value near 0.5 means no correlation between choice and spike-firing activity is apparent (i.e., chance level).

## Auditory brainstem response (ABR) recording

We recorded ABRs from a majority of our animals (Ctl: n = 9; HL: n = 8) to compare peripheral status and auditory sensitivity upstream of ACx (e.g., brainstem function) between Ctl and HL groups. ABR procedures are identical to those of previous studies from our laboratory (*Rosen et al., 2012*; *Caras and Sanes, 2015*). Briefly, once animals accomplished all behavioral training and testing stages, they were anesthetized with intraperitoneal injections of ketamine (30 mg/kp; Bioniche Pharma) and pentobarbital (50 mg/kg; Sigma-Aldrich) and placed in a small sound booth (Industrial Acoustics). Animal internal temperature was maintained at 37°C by a digitally controlled heating blanket. Pin electrodes were inserted subcutaneously at the vertex of the skull (positive electrode) and caudal to the right pinna (inverting electrode). A ground pin electrode was inserted into the left leg. A preamplifier (P55; Grass Technologies) amplified (10,000 gain) voltage responses and signals were bandpass filtered between 0.3–3 kHz with an additional gain of 14 dB (Brownless Precision

Model 440 amplifier, AutoMate Scientific). Signals were digitized at 24.4 kHz sampling rate (RZ6, Tucker Davis Technologies) and acquired on a personal Microsoft Windows computer.

Stimulus generation and data acquisition were controlled by custom Python scripts (provided by Brandon Warren and Edwin Rubel, University of Washington, Seattle). We presented two types of acoustic stimuli that were delivered from a calibrated free-field speaker positioned 26 cm above the animal's head. One type consisted of 100 µs clicks that were repeated 23/s at alternating polarity and in 10 dB descending steps until the full ABR disappeared, at which point the intensity was increased by 5 dB SPL until a reliable response was observed (online visually detectable N1 potential). The lowest sound level (dB SPL) at which a reliable response was seen was taken as threshold. Traces were averaged across 250–500 sweeps.

We also delivered 50–100 ms broadband AM noise (0.1–20 kHz) with a 5 ms cosine ramp at onset and offset across behaviorally tested rates (64, 128, 256, and 512 Hz) and modulation depths (−15 to 0 dB rel. 100% depth). An unmodulated noise signal was also presented. AM and unmodulated noise stimuli were presented 6.6/s and at +30 dB above click threshold (~45–50 dB SPL for Ctl;~85–90 dB for HL). Auditory-evoked potentials to these stimuli were averaged across 1,000 repetitions and referred to as envelope following responses (EFRs) to signify their relationship with the temporal modulation envelope (*Parthasarathy and Bartlett, 2012*; *Parthasarathy et al., 2014*; *Parthasarathy et al., 2016*). We quantified the degree of phase-locking by performing fast Fourier transforms (FFTs) on the waveforms between +10 ms stimulus onset to −10 ms stimulus offset. The peak FFT amplitude is labeled the maximum energy at the presented modulation frequency or within 3 Hz above and below it (*Parthasarathy et al., 2010*; *Parthasarathy and Bartlett, 2011*; *Parthasarathy and Bartlett, 2012*; *Parthasarathy et al., 2014*; *Parthasarathy et al., 2016*). The peak FFT amplitudes evoked across AM depths were fitted with exponential functions. The 'noise floor' was calculated from the unmodulated noise condition and the threshold for each AM rate was defined as the lowest modulation depth with a peak FFT amplitude that is twice as high as the noise floor.

In addition to the initial electrode configurations described above (hereafter referred to as 'Configuration 2'), we also utilized a separate configuration where the positive pin electrode was placed horizontally, along the interaural line, above the standard location of the inferior colliculus (hereafter referred to as 'Configuration 1'). 'Configuration 2' emphasizes waves I and III of the ABR with robust EFRs to AM rates > 100 Hz, whereas 'Configuration 1' emphasizes waves IV and V with robust EFRs to AM rates < 100 Hz. It is suggested that waves I to III are generated by portions of the auditory nerve (*Sohmer et al., 1974*; *Buchwald and Huang, 1975*; *Starr and Hamilton, 1976*; *Achor and Starr, 1980*) through the cochlear nucleus (*Møller and Jannetta, 1982*; *Melcher and Kiang, 1996*; *Melcher et al., 1996*). In addition, waves IV and V reflect activity from multiple neural generators that include the superior olivary complex, lateral lemniscus tract, and lateroventral inferior colliculus (*Hashimoto et al., 1981*; *Møller and Jannetta, 1982*; *Møller and Jannetta, 1983*; *Funai and Funasaka, 1983*; *Kaga et al., 1997*). With regards to temporal processing across the ascending auditory system, previous studies show that the two distinct pin electrode configurations for recording EFRs can represent discrete neural generators (*Parthasarathy and Bartlett, 2012*; *Parthasarathy et al., 2014*). Specifically, the more standard electrode setup ('Configuration 2') exhibits more robust EFRs at faster AM rates (>100 Hz), which may emphasize more caudal brainstem structures through the auditory nerve. In addition, the electrode set up described as 'Configuration 1' exhibits more robust responses at slower modulation rates (<100 Hz) and emphasizes more rostral auditory nuclei that include the inferior colliculus, suggesting that utilizing both Configuration 1 and 2 electrode setups is important for comparing multiple levels of temporal processing along the rostral to caudal auditory pathway. Furthermore, two studies have demonstrated that scalp-recorded potentials to fast AM rates (i.e., greater than ~80 Hz) in unanesthetized animals (cat and rabbit) are produced by neural generators at or below the auditory midbrain (*Kiren et al., 1994*; *Kuwada et al., 2002*). Here, we utilized EFR measures as an assay of auditory temporal processing with the assumption that the EFR corresponds to a measure of population temporal fidelity up to the level of the brainstem.

## Statistical procedures

Statistical analyses and procedures were implemented in JMP 13.2.0 (SAS) or custom-written MATLAB scripts (The MathWorks) that incorporated the MATLAB Statistics Toolbox. For normally distributed data (as assessed by the Lilliefors test), data are reported as mean ±SEM unless otherwise

stated. When data were not normally distributed, either the non-parametric Kolmogorov-Smirnov or Wilcoxon Rank Sum test was used when appropriate. For post-hoc multiple comparisons analyses, alpha values were Holm-Bonferroni-corrected. When violations of sphericity were present, p values and degrees of freedom were Greenhouse-Geisser corrected. To test for statistically significant between-group differences, while accounting for differences in the number of units recorded from each animal (multi-units: 80–285, single-units: 16–50), we compared group differences on a subsample of the population data set with 10,000 bootstrapped iterations. During each iteration, we randomly sampled 30 units from each animal (without replacement) and calculated the average difference between hearing status group (Ctl versus HL). After 10,000 iterations, empirical two-tailed 98.75% confidence intervals (CIs) were calculated from the distribution. A statistically significant difference between groups tested at p<0.05 was determined if zero fell outside the 98.75% CIs.

## Acknowledgements

The work is supported by National Institute of Health Grants R01 DC014656 (DHS), T32 MH019524 (JDY), and F32 DC016508 (JDY). We thank Dr. Edward Bartlett for expert advice regarding envelope following response measures from the auditory brainstem. We also thank Melissa Caras, Kristina Penikis, and Kelsey Anbuhl for helpful discussions and editorial comments on the manuscript.

## Additional information

### Funding

| Funder | Grant reference number | Author |
|---|---|---|
| National Institute on Deafness and Other Communication Disorders | F32 DC016508 | Justin D Yao |
| National Institute on Deafness and Other Communication Disorders | T32 MH019524 | Justin D Yao |
| National Institute on Deafness and Other Communication Disorders | R01 DC014656 | Dan H Sanes |

The funders had no role in study design, data collection and interpretation, or the decision to submit the work for publication.

### Author contributions

Justin D Yao, Conceptualization, Data curation, Formal analysis, Funding acquisition, Investigation, Visualization, Methodology, Writing—original draft, Writing—review and editing; Dan H Sanes, Conceptualization, Supervision, Funding acquisition, Visualization, Methodology, Writing—original draft, Project administration, Writing—review and editing

### Author ORCIDs

Justin D Yao (iD) http://orcid.org/0000-0001-8762-9044
Dan H Sanes (iD) http://orcid.org/0000-0002-3783-6165

### Ethics

Animal experimentation: All procedures of this study were approved by the Institutional Animal Care and Use Committee at New York University and followed guidelines established by the National Institutes of Health for the care and use of laboratory animals. All conductive hearing loss surgeries were performed under a surgical level of anesthesia induced with methoxyflurane. All auditory brainstem response recordings were performed under ketamine and pentobarbital. All electrode implant surgeries were performed under isoflurane/O2. Every effort was made to minimize suffering.

Decision letter and Author response
Decision letter https://doi.org/10.7554/eLife.33891.015
Author response https://doi.org/10.7554/eLife.33891.016

## Additional files

### Supplementary files
• Transparent reporting form
DOI: https://doi.org/10.7554/eLife.33891.012

### Data availability
MATLAB files and code are available at the New York University Box (https://nyu.box.com/v/Yao-Sanes-eLife-2018).

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
