## [Decision Letter]

Thank you for submitting your article "Developmental deprivation-induced perceptual and cortical processing deficits in awake-behaving animals" for consideration by *eLife*. Your article has been reviewed by three peer reviewers, and the evaluation has been overseen by Eve Marder as the Senior/Reviewing Editor. The reviewers have opted to remain anonymous.

There was extensive discussion among the reviewers about how conclusive they felt these observations to be, and there were a variety of graded opinions about how to deal with these diverse takes on the manuscript, and whether rewriting would be able to deal with them conclusively. Therefore, at odds with the usual *eLife* practice, we are including all three of the initial reviews in entirety. In your revision, please provide the best and most comprehensive replies to the issues raised in the reviews. Many of the concerns will require more detail, and to some degree our final assessment will depend on how well you are able to deal with the concerns, although all of the reviewers find the work addresses an important set of problems.

Reviewer #1:

This manuscript reports, for normal hearing animals and in animals with a developmental conductive hearing loss, that behavioral performance in an amplitude-modulation (AM) detection task is reflected in auditory cortical but not brainstem population activity. A reduction of the detection ability for high AM frequencies following a moderate, conductive threshold shift is documented.

Overall, this a technically well-executed study with careful controls and sensible analytical approaches. Major assets are that cortical recordings were made while animals performed the task. The observation that behavioral AM performance in Gerbils could be traced to cortical population firing rate changes in normal hearing animals confirms previous reports in this and other species. The fact that this was also true for animals with a substantial conductive hearing loss is certainly novel, even if it only corroborates that a general peripheral threshold shift does not seem to alter how behavior is reflected in sensory cortical activity. The observed reduction of AM-depth detection at high AM frequencies provides behavioral and electrophysiological evidence that parallels observations in humans and supports the hypothesis that chronic conductive loss, likely via reduced inhibition, generates a wider range of temporal response variability, constraining central auditory AM encoding abilities.

A couple of issues, however, diminish the potential interpretations and impact of this approach.

The issue with most attempts to compare psychometric and neurometric thresholds is that there seem to be too many free parameters on the physiology side. What if the decoding model is suboptimal, e.g., missing relevant temporal patterns? What if the assumptions about the decoding model are not biologically realizable? What if there are different subpopulations and the relevant population is undersampled? What if irrelevant neurons are oversampled, corrupting the representation?

The reliance on rate measures is a bit puzzling as work in monkey has shown that primary cortex does utilize temporal response aspects and, in addition, that rate measures may be correlated with performance but not always with a monotonic increase in FR with better performance. How is a population approach, as used here, affected by these different codes and neuronal stimulus dependencies?

Another aspect, previously addressed by the authors and others, is that top-down influences shape learning as well as neuronal properties during performance. Is it conceivable that the observed effects are more a reflection of the recurrent decision process than the encoding/decoding contributions of AC itself?

The claim that the behavioral performance reduction is not reflected in brainstem activity is intriguing but perhaps premature. The main question is what the population activity observed with ABRs and EFRs actually tells us about the local activity of several stations. The summated activity is certainly significantly affected by the degree of synchrony within and across those stations, potentially obscuring subpopulation contributions and firing rate modulations which were the key to the observations in AC. Also, the brainstem responses were not obtained while the animals where performing the task. This may not be necessary to make the point that brainstem activity is seemingly not affected by the hearing loss but further dissociates the two main parts of the study.

Reviewer #2:

This is a great paper, where the authors investigate the effect of a conductive hearing loss on brainstem processing, cortical responses, and perceptual deficits. They focus on two primary results. The first is that conductive hearing loss leads to a reduction in performance on an auditory detection task, with no alteration in brainstem temporal processing. The second is that perceptual deficits were associated with a degraded population code.

Overall, I think it's a lovely paper, and worthy of publication in *eLife*. I do have some specific comments that I feel will improve the quality of the paper.

1) There are a number of different factors that could alter sound responses in this kind of task. Could you please provide details as to the size of the behavioral apparatus, and comment on the following issues? Specifically since there is only one speaker, the sound path to the animals ear will change as the animal navigates. What is the natural variability in neural responses that you would expect from roaming in such an environment (compared to what you observed in the group differences)? Locomotion effects might also change both firing rate and response variability. Were there systematic differences in the way in which the different groups moved around the arena?

2) "ABR thresholds to clicks, which is a general measure of hearing threshold…". And all other references to ABR's reflecting hearing. There has been work showing that ABRs do not really measure "hearing", they measure brainstem function. Chambers et al. (2016) have shown that lesioning >95% of cochlear afferent synapses can virtually eliminate the ABR but leave tone detection behavior completely normal. It would be nice to make this point clear in the paper.

3) Statistics used should be related to the way the data is presented. As an example, the data in Figure 6D shows median value for the two groups, but the statistical test used does not test for a difference in median. The KS test tests whether the underlying probability distributions differ, and it does this by calculating the KS statistic from the CDF's (so this test is more appropriate when directly comparing CDFs in Figure 6). If the point is to be made that there is a difference in median, then perhaps something like a Wilcoxon Rank Sum test would be a better choice.

4) Population decoding performance normally scales with the number of neurons in the population. Is this the case in your data? And, if so, can you control for this when comparing decoder accuracy between groups of different sizes?

5) You've shown an interesting change in variability between groups, but I wonder whether there may also be differences in co-variability between groups. Have you looked at noise correlations at all (between simultaneously recorded units)? If there was a difference between groups, then the importance of noise correlations could also be studied with the population decoder by shuffling the trial order.

6) With regards to Figure 6A. This figure is showing CDFs of firing rates between groups of animals. I don't think you mention anywhere what the individual animal N is. Could you please provide evidence that this between group difference isn't being driven by a particular (or small group) of animals. This could be achieved (for example) by constructing a bootstrapped statistical test, whereby the same N was drawn repeatedly from each animal. Could you also please show the data that these differences (in FR and CV) holds up for single-units?

Reviewer #3:

In this study, the authors investigate the behavioural and neurophysiological consequences of a permanent developmental conductive hearing loss in gerbils. Unlike previous studies of this topic, the authors are able to record from auditory cortical neurons whilst the animals perform a behavioural task. They find that animals reared with hearing loss are less sensitive to amplitude modulation of sounds, but only for faster modulation rates. A population decoder applied to cortical neurons shows a broadly similar deficit. The authors argue that this is because the responses of cortical neurons are more variable across trials in animals reared with hearing loss. Auditory brainstem responses (specifically the envelope following response) do not show a deficit in animals reared with hearing loss. The authors therefore conclude that the behavioural deficit in temporal processing emerges above the level of the brainstem, perhaps even in the cortex itself.

Overall, the authors should be commended for tackling an important topic using an ambitious and novel methodological approach that produces an interesting set of results. However, there are a couple of areas that could be written more clearly or need additional detail. There are also a number of important differences between the behavioural and neurophysiological data, and it would be helpful to identify these clearly and discuss their implications at greater length.

The methods used for population decoding need to be explained more clearly and in greater detail.

Although the population decoder data share certain features with the behavioural data, there are also a number of important differences. These need to be clarified and discussed at greater length. Estimating thresholds from non-monotonic data, or data that do not cross threshold, also poses a considerable problem that needs to be discussed. Without a consideration of these issues, it is difficult to fully assess the degree of similarity between the behavioural and neurophysiological data.

It would also be helpful to discuss the results in the context of previous anaesthetized work.

---

## [Author Response]

There was extensive discussion among the reviewers about how conclusive they felt these observations to be, and there were a variety of graded opinions about how to deal with these diverse takes on the manuscript, and whether rewriting would be able to deal with them conclusively. Therefore, at odds with the usual eLife practice, we are including all three of the initial reviews in entirety. In your revision, please provide the best and most comprehensive replies to the issues raised in the reviews. Many of the concerns will require more detail, and to some degree our final assessment will depend on how well you are able to deal with the concerns, although all of the reviewers find the work addresses an important set of problems.

To address the issue of conclusiveness, we have: (1) extended the population decoder analyses to address many of the reviewers’ concerns (e.g., a subpopulation of auditory cortex neurons does optimally encode amplitude modulation depth; see Figure 7B and C and new Figure 8); (2) provided new data indicating that the animal’s motor behavior is consistent across hearing status groups, and that roaming does not introduce significant variability to the recorded responses; (3) provided a statistical test to show that the most interesting finding, an increase in spike firing variability in the HL-reared group, was not being driven by an individual animal.

Reviewer #1:[…] A couple of issues, however, diminish the potential interpretations and impact of this approach.The issue with most attempts to compare psychometric and neurometric thresholds is that there seem to be too many free parameters on the physiology side. What if the decoding model is suboptimal, e.g., missing relevant temporal patterns? What if the assumptions about the decoding model are not biologically realizable? What if there are different subpopulations and the relevant population is undersampled? What if irrelevant neurons are oversampled, corrupting the representation?

These are all logical questions, and we can address some of them empirically. However, we acknowledge that external validation does not yet exist for decoding models.

To evaluate whether the decoding model is missing relevant temporal patterns, we calculated how well the ACx population could differentiate between Go (100% AM Depth) versus Nogo (unmodulated) signals across a broad range of bin sizes. Figure 7B (new) shows that the best readout performance (maximum *d’*) from our decoding analysis peaked at a bin size of 1 ms for all tested AM rates, and in both hearing status groups. In addition, we examined population decoding performance as a function of population size. For this analysis, a resampling procedure was applied to randomly select a subpopulation of units (10-100% of total units) across 250 iterations. During each iteration of the resampling procedure, a new subpopulation of units was randomly selected (without replacement) from all units prior to the decoding readout procedure. Average performance and variability ( ± 1 SD) were calculated across 250 iterations. Figure 7C displays the decoding readout performance as a function of the number of units for each AM rate tested. In general, the inclusion of all units produced optimal decoding performance.

To evaluate the impact of an ACx subpopulation, we calculated the performance of neurons that increased their firing rate as a function of AM depth versus those that decreased their FR. In the original manuscript, we reported that FR increased in 51.7% of units and decreased in 48.3% (examples in Figure 6B). We now formalize this distinction by computing a “monotonicity index” (MI; de la Rocha et al., 2008; Sadagopan and Wang, 2008; Middlebrooks, 2008a,b; Watkins and Barbour, 2011). Specifically, the MI is the FR at 100% modulation depth expressed as a percentage of the maximum FR observed across all depths. Indices of 100% indicate units that display a monotonic increase in FR with increasing modulation depth, with smaller indices indicating nonmonotonic responses (distribution shown in Figure 8A). Thus, we compared decoder readout performance between nonmonotonic versus monotonic units, using MI values ≤ 50% as a criterion for nonmonotonicity. A sufficient number of nonmonotonic neurons were available at 128 and 256 Hz conditions (Figure 8B) to perform this analysis. Decoder performance was superior for nonmonotonic units across most AM depths at 128 and 256 Hz, both from Ctl and HL groups. This result suggests that optimal decoding of fast AM information could rely on a subpopulation of ACx neurons. That said, the inclusion of all units more closely resembles psychometric performance as suggested by previous studies (e.g., Shadlen et al., 1998).

The reliance on rate measures is a bit puzzling as work in monkey has shown that primary cortex does utilize temporal response aspects and, in addition, that rate measures may be correlated with performance but not always with a monotonic increase in FR with better performance. How is a population approach, as used here, affected by these different codes and neuronal stimulus dependencies?

Although auditory cortex neurons display little or no phase locking to the envelope for AM rates >100 Hz (e.g., Lu and Wang, 2000; Malone et al. 2007), it is possible that fine-grained temporal structure could carry information. This is addressed, in part, with the new analysis of bin size (above). Similarly, an evaluation of monotonic increases in FR is discussed above.

Another aspect, previously addressed by the authors and others, is that top-down influences shape learning as well as neuronal properties during performance. Is it conceivable that the observed effects are more a reflection of the recurrent decision process than the encoding/decoding contributions of AC itself?

The reviewer raises an excellent point. Many studies demonstrate that engagement in an auditory-guided task modulates ACx response properties (Fritz et al., 2003, 2005; Otazu et al., 2009; Lee and Middlebrooks, 2011; Niwa and Sutter, 2012; Schneider et al., 2014; Zhou et al., 2014; Slee and David, 2015; von Trapp et al., 2016). That was actually the primary motivation for the present study, since previous studies of this sort on developmental plasticity have compared separate groups of animals for measures of physiology and behavior.

The most appropriate paradigm to test for potential top-down influences would be to assess neuronal response properties evoked by the same auditory stimuli across separate conditions of on-task (engaged) versus off-task (disengaged). In fact, we are currently conducting a separate set of experiments where ACx responses from Ctl and HL gerbils are obtained during task engaged and disengaged conditions, using an AM discrimination task.

The claim that the behavioral performance reduction is not reflected in brainstem activity is intriguing but perhaps premature. The main question is what the population activity observed with ABRs and EFRs actually tells us about the local activity of several stations. The summated activity is certainly significantly affected by the degree of synchrony within and across those stations, potentially obscuring subpopulation contributions and firing rate modulations which were the key to the observations in AC. Also, the brainstem responses were not obtained while the animals where performing the task. This may not be necessary to make the point that brainstem activity is seemingly not affected by the hearing loss but further dissociates the two main parts of the study.

One possibility that we had not considered is that a subpopulation of brainstem neurons is impaired following developmental HL, yet this impairment is masked during a population level assessment of function, such as ABRs and EFRs. There is precedent for this sort of null finding in the literature: tone detection thresholds that are assessed with ABRs following noise induced hearing loss can appear normal even when some cochlear synapses have been lost (Kujawa and Liberman, 2009). To address this issue, we made two changes to the manuscript: (1) We have backed off from the strong assertion that the brainstem is not impaired by HL and, instead, state throughout the manuscript that our recordings do not reveal an impairment. This is accurate and leaves open the possibility that future awake-behaving studies that record from auditory brainstem structures may reveal interesting changes. (2) We have added text to the Discussion that explicitly acknowledges this possibility. In the same passage, we acknowledge that inferior colliculus neurons are modulated by task-engagement (Slee and David, 2015), leaving open the possibility that EFRs obtained during performance of the AM detection task could have been differentially modulated in Ctl and HL animals.

Reviewer #2:[…] Overall, I think it's a lovely paper, and worthy of publication in eLife. I do have some specific comments that I feel will improve the quality of the paper.1) There are a number of different factors that could alter sound responses in this kind of task. Could you please provide details as to the size of the behavioral apparatus, and comment on the following issues?

The following information was added to the Materials and methods section:

“Adult gerbils were placed in a plastic test cage (0.25 x 0.25 x 0.4 m) in a sound-attenuating booth (Industrial Acoustics; internal dimensions: 2.2 x 2 x 2 m) and observed via a closed-circuit monitor.”

Specifically since there is only one speaker, the sound path to the animals ear will change as the animal navigates. What is the natural variability in neural responses that you would expect from roaming in such an environment (compared to what you observed in the group differences)?

Although the speaker position (above the test cage) was chosen to minimize interaural cues, it is possible that we have underestimated this effect. Using a set of animals that are currently performing an AM rate discrimination task, we ran a set of pilot experiments in which the location of the nose poke and reward apparatus (food tray in this particular case) were switched halfway through the session. This manipulation effectively alters head position by 180° at the beginning (nosepoke) and end (waterspout) of each Go trial, and permits us to examine potential differences in neural responses that arise due to the animal’s navigational path. Author response image 1 shows preliminary data from two ACx units that suggest no difference in neural responses when head position was altered.

**Author response image 1. respfig1:** Preliminary assessment of head position. No difference in neural responses for separate nosepoke and reward apparatus locations was observed in these two example ACx units.

Locomotion effects might also change both firing rate and response variability. Were there systematic differences in the way in which the different groups moved around the arena?

To address this question, we compared response latencies between Ctl versus HL animals for Go (approached water spout) and Nogo (repoked) responses across each tested AM rate. No main effect of hearing status (Ctl versus HL) was observed for trials when animals approached the water spout (two-way mixed model ANOVA; F_(1,5266)_ = 3.97, p > 0.05) or for trials when animals repoked (two-way mixed model ANOVA; F_(1,2514)_ = 0.18, p > 0.05). This suggests that there were no systematic differences in motor behavior between Ctl and HL animals. This information is included in the revised manuscript.

2) "ABR thresholds to clicks, which is a general measure of hearing threshold…". And all other references to ABR's reflecting hearing. There has been work showing that ABRs do not really measure "hearing", they measure brainstem function. Chambers et al. (2016) have shown that lesioning >95% of cochlear afferent synapses can virtually eliminate the ABR but leave tone detection behavior completely normal. It would be nice to make this point clear in the paper.

We now make it clear that we used the ABR as a measure of brainstem function, and primarily to validate the sound level settings used for behavioral experiments on Ctl and HL animals.

3) Statistics used should be related to the way the data is presented. As an example, the data in Figure 6D shows median value for the two groups, but the statistical test used does not test for a difference in median. The KS test tests whether the underlying probability distributions differ, and it does this by calculating the KS statistic from the CDF's (so this test is more appropriate when directly comparing CDFs in Figure 6). If the point is to be made that there is a difference in median, then perhaps something like a Wilcoxon Rank Sum test would be a better choice.

We now use the Wilcoxon Rank Sum test when appropriate.

4) Population decoding performance normally scales with the number of neurons in the population. Is this the case in your data? And, if so, can you control for this when comparing decoder accuracy between groups of different sizes?

We have added this analysis. Figure 7C displays the decoder performance as a function of the number of units for each AM rate tested from both groups. To account for differences in sub-group sizes, we applied a bootstrapped resampling procedure to randomly select a subpopulation of units. Specifically, on each iteration of the resampling, a new subpopulation of units was randomly selected (without replacement) from all units, and trials were randomly assigned for training and testing (without replacement).

5) You've shown an interesting change in variability between groups, but I wonder whether there may also be differences in co-variability between groups. Have you looked at noise correlations at all (between simultaneously recorded units)? If there was a difference between groups, then the importance of noise correlations could also be studied with the population decoder by shuffling the trial order.

We are very interested in this type of analysis. However, we did not acquire a sufficient number single units within individual sessions. Since this is a limitation of our current recording configuration (16 channel array), we have recently obtained a headstage that permits recordings from up to 64 channels, and this should permit us to analyze noise correlations (Stringer et al., 2016).

6) With regards to Figure 6A. This figure is showing CDFs of firing rates between groups of animals. I don't think you mention anywhere what the individual animal N is. Could you please provide evidence that this between group difference isn't being driven by a particular (or small group) of animals. This could be achieved (for example) by constructing a bootstrapped statistical test, whereby the same N was drawn repeatedly from each animal. Could you also please show the data that these differences (in FR and CV) holds up for single-units?

We now include the number of animals (N_gerbils_, Ctl = 3; HL = 3) and units (N_units_, Ctl = 484; HL = 331) for which recordings were obtained during psychophysical testing. For single units, we found that driven and spontaneous FRs were significantly greater among Ctl units (two-sample Kolmogorov-Smirnov test; Driven FR: K = 0.32, p < 0.0005; Spon FR: K = 0.39, p < 0.0001). In addition, CV was significantly lower among Ctl single units (two-sample Kolomogorov-Smirnov test; K = 0.25, p < 0.01).

The number of units recorded from each individual animal ranged from 80-285 (single units: 16-50). To account for differences in the total number of units recorded from any particular animal, we assessed group differences in FR, Spont FR, and CV with a bootstrapped statistical test (see Materials and methods). Using this more conservative approach, we found no significant difference in FR between Ctl versus HL units (confidence interval (CI) = [-3.09 2.99], p > 0.05). However, Spont FR remained significantly higher among Ctl units (CI = [1.59 9.04], p < 0.05) and CV remained significantly lower among Ctl units (CI = [-0.14 -0.04], p < 0.05). In addition, we analyzed single- and multi-unit data separately with the same bootstrapped statistical procedure and found that results were consistent across single- and multi-unit populations (Figure 6—figure supplement 1). This information has been added to the revised manuscript.

Reviewer #3:[…] Overall, the authors should be commended for tackling an important topic using an ambitious and novel methodological approach that produces an interesting set of results. However, there are a couple of areas that could be written more clearly or need additional detail. There are also a number of important differences between the behavioural and neurophysiological data, and it would be helpful to identify these clearly and discuss their implications at greater length.The methods used for population decoding need to be explained more clearly and in greater detail.

We now provide further descriptions for clarity.

Although the population decoder data share certain features with the behavioural data, there are also a number of important differences. These need to be clarified and discussed at greater length. Estimating thresholds from non-monotonic data, or data that do not cross threshold, also poses a considerable problem that needs to be discussed. Without a consideration of these issues, it is difficult to fully assess the degree of similarity between the behavioural and neurophysiological data.

The reviewer makes an excellent point. We extended our descriptions of the population decoder analysis and additionally include comparisons across subpopulations of nonmonotonic versus monotonic units.

It would also be helpful to discuss the results in the context of previous anaesthetized work.

We now include this comparison in the Discussion section.